# Deep Generative Wasserstein Gradient Flows

## Abstract

Deep generative modeling is a rapidly-advancing field with a wealth of modeling choices developed in the past decade. Amongst them, Wasserstein gradient flows (WGF) are a powerful and theoretically rich class of methods. However, their applications to high-dimensional distributions remain relatively underexplored. In this paper, we present Deep Generative Wasserstein Gradient Flows (DGGF), which constructs a WGF minimizing the entropy-regularized $f$-divergence between two distributions. We demonstrate how to train a deep density ratio estimator that is required for the WGF and apply it to the task of generative modeling. Experiments demonstrate that DGGF is able to synthesize high-fidelity images of resolutions up to $128 \times 128$, directly in data space. We demonstrate that DGGF has an interpretable diagnostic of sample quality by naturally estimating the KL divergence throughout the gradient flow. Finally, we show DGGF's modularity by composition with external density ratio estimators for conditional generation, as well as for unpaired image-to-image translation without modifications to the underlying framework.

## 1 Introduction

Gradient flow methods are a powerful and general class of techniques with diverse applications ranging from physics (Carrillo et al., 2019; Adams et al., 2011) and sampling (Bernton, 2018) to neural network optimization (Chizat & Bach, 2018) and reinforcement learning (Richemond & Maginnis, 2017; Zhang et al., 2018). In particular, Wasserstein gradient flow (WGF) methods are a popular specialization that model the gradient dynamics on the space of probability measures with respect to the Wasserstein metric; these methods aim to construct the *optimal* path between two probability measures — a source distribution $q(\mathbf{x})$ and a target distribution $p(\mathbf{x})$ — where the notion of optimality refers to the path of steepest descent in Wasserstein space.

The freedom in choosing $q(\mathbf{x})$ and $p(\mathbf{x})$ when constructing the WGF makes the framework a natural fit for a variety of generative modeling tasks. For data synthesis, we choose $q(\mathbf{x})$ to be a simple distribution easy to draw samples from (e.g., Gaussian), and $p(\mathbf{x})$ to be a complex distribution which we would like to learn (e.g., the distribution of natural images). The WGF then constructs the optimal path from the simple distribution to synthesize data resembling that from the complex distribution. Furthermore, we could choose both $p(\mathbf{x})$ and $q(\mathbf{x})$ to be distributions from different domains of the same modality (e.g., images from separate domains). The WGF then naturally performs domain translation.

However, despite this fit and the wealth of theoretical work established over the past decades (Ambrosio et al., 2005; Santambrogio, 2017), applications of WGFs to generative modeling of high-dimensional distributions remain under-explored and limited. A key difficulty is that the 2-Wasserstein distance and divergence functionals are generally intractable. Existing works rely on complex optimization schemes with constraints that contribute to model complexity, such as approximations of the 2-Wasserstein distance with input convex neural networks (Mokrov et al., 2021), dual variational optimization schemes with the Fenchel conjugate (Fan et al., 2021) or adopting a particle simulation approach, but amortizing sample generation to auxiliary generators (Gao et al., 2019; 2022).

In this work, we take a step towards resolving the shortcomings of WGF methods for deep generative modeling. We propose Deep Generative Wasserstein Gradient Flows (DGGF), which is formulated using the gradient flow of entropy-regularized $f$-divergences (Fig. 1). As this formulation involves

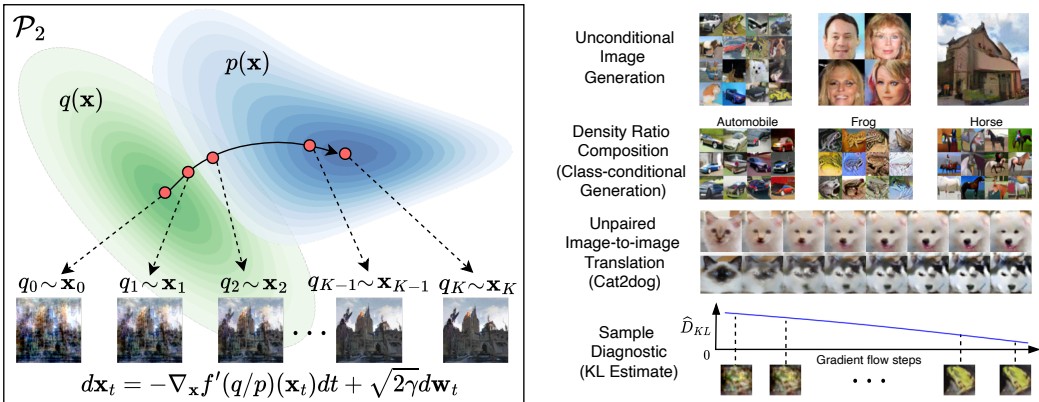

Figure 1: Left: illustration of the generative gradient flow process using DGGF. The evolution of the gradient flow is governed by the SDE shown in the figure. We visualize intermediate samples of the LSUN Church dataset. Right: visualization of application domains of DGGF. At its core, DGGF is able to perform high-fidelity unconditional image generation. The unconditional model can be used for class-conditional generation via density ratio composition with external pretrained classifiers. Additionally, DGGF is able to perform unpaired image-to-image translation with no modifications needed to the framework. Finally, DGGF possesses an innate sample diagnostic by estimating the KL divergence over the flow, which decreases as sample quality is improved over the flow.

density ratio estimation, we introduce a novel algorithm for training deep density ratio estimators and show experimentally for the first time that gradient flow methods can scale to image dimensions as high as $128 \times 128$. Our gradient flow is formulated entirely in the data space, with no need for additional generator networks. The density ratio formulation allows DGGF to be composed with external density ratio estimators, which we show allows us to utilize pretrained external classifiers for class-conditional generation. In addition, we demonstrate that DGGF can be viewed as estimating the KL divergence of samples over the flow, providing it with an innate diagnostic to evaluate sample quality that also enhances model interpretability. We also show a simple technique of leveraging data-dependent priors to boost generative performance. Finally, by leveraging the freedom of choosing the source and target distributions, we show DGGF can be applied to unpaired image-to-image translation with no modifications to the framework.

## 2 BACKGROUND

In the following, we give a brief overview of gradient flows and density ratio estimation. For a more comprehensive introduction to gradient flows, please refer to Santambrogio (2017). A thorough overview of density ratio estimation can be found in Sugiyama et al. (2012a).

**Wasserstein Gradient Flows.** To motivate the concept of gradient flows, we consider Euclidean space equipped with the familiar $L_2$ distance metric $(\mathcal{X}, \|\cdot\|_2)$. Given a function $F : \mathcal{X} \to \mathbb{R}$, the curve $\{\mathbf{x}(t)\}_{t \in \mathbb{R}^+}$ that follows the direction of steepest descent is called the *gradient flow* of $F$:

$$\mathbf{x}'(t) = -\nabla F(\mathbf{x}(t)). \tag{1}$$

In generative modeling, we are interested in sampling from the *probability distribution* of a given dataset. Hence, instead of Euclidean space, we consider the space of *probability measures* with finite second moments equipped with the 2-Wasserstein metric $(\mathcal{P}_2(\Omega), \mathcal{W}_2)$. Given a functional $\mathcal{F} : \mathcal{P}_2(\Omega) \to \mathbb{R}$ in the 2-Wasserstein space, the gradient flow of $\mathcal{F}$ is the steepest descent curve of $\mathcal{F}$. We call such curves Wasserstein gradient flows (WGF).

**Density Ratio Estimation via Bregman Divergence.** Let $q(\mathbf{x})$ and $p(\mathbf{x})$ be two distributions over $\mathcal{X} \in \mathbb{R}^d$ where we have access to i.i.d samples $\mathbf{x}_q \sim q(\mathbf{x})$ and $\mathbf{x}_p \sim p(\mathbf{x})$. The goal of density ratio estimation (DRE) is to estimate the true density ratio $r^*(\mathbf{x}) = \frac{q(\mathbf{x})}{p(\mathbf{x})}$ based on samples $\mathbf{x}_q$ and $\mathbf{x}_p$.

We will focus on density ratio fitting under the Bregman divergence (BD), which is a framework that unifies many existing DRE techniques (Sugiyama et al., 2012a;b). Let $g : \mathbb{R}^+ \to \mathbb{R}$ be a twice continuously differentiable convex function with a bounded derivative. The BD seeks to quantify the discrepancy between the estimated density ratio $r_\theta$ and the true density ratio $r^*$:

$$BD_g(r^*||r_\theta) = \mathbb{E}_{p(\mathbf{x})}[g(r^*(\mathbf{x})) - g(r_\theta(\mathbf{x})) + \partial g(r_\theta(\mathbf{x}))r_\theta(\mathbf{x})] - \mathbb{E}_{q(\mathbf{x})}[\partial g(r_\theta(\mathbf{x}))]. \quad (2)$$

As we only have access to samples, we estimate the expectations in Eq. 2 using Monte Carlo estimates:

$$BD_g(r_\theta) = \frac{1}{N}\sum_{n=1}^{N}[\partial g(r_\theta(\mathbf{x}_p^{(n)}))r_\theta(\mathbf{x}_p^{(n)}) - g(r_\theta(\mathbf{x}_p^{(n)}))] - \frac{1}{N}\sum_{n=1}^{N}[\partial g(r_\theta(\mathbf{x}_q^{(n)}))], \quad (3)$$

where we drop the term $\mathbb{E}_p[g(r^*(\mathbf{x}))]$ as it does not depend on the model $r_\theta$ during optimization. The minimizer of Eq. 3, which we denote $\theta^*$, satifies $r_{\theta^*}(x) = r^*(\mathbf{x}) = q(\mathbf{x})/p(\mathbf{x})$. For ease of notation, we will use the hatted symbol $\widehat{\mathbb{E}}_p$ to refer to Monte Carlo estimates.

## 3 GENERATIVE MODELING WITH WASSERSTEIN GRADIENT FLOWS

This section describes our key contribution: Deep Generative Wasserstein Gradient Flows (DGGF), in which we show how to train a deep density ratio estimator for high-fidelity generative modeling. Let $q(\mathbf{x})$ and $p(\mathbf{x})$ be two distributions over $\mathcal{X} \in \mathbb{R}^d$. Assume that $p(\mathbf{x})$ is the target distribution that we wish to learn. We choose $q(\mathbf{x})$ to be a known distribution that we can sample from, such as a uniform or Gaussian distribution. Our goal is to construct a WGF starting from $q(\mathbf{x})$, such that trajectories along the flow decrease some notion of distance between $q(\mathbf{x})$ and $p(\mathbf{x})$. This will allow us to flow samples from $q(\mathbf{x})$ to $p(\mathbf{x})$; in our case, when $q(\mathbf{x})$ is a simple prior and $p(\mathbf{x})$ is the distribution of natural images, we can perform generative modeling of images.

Formally, the functional $\mathcal{F} : \mathcal{P}_2(\Omega) \to \mathbb{R}$ encodes the aforementioned notion of distance between $q(\mathbf{x})$ and $p(\mathbf{x})$. Following Ansari et al. (2021), we choose $\mathcal{F}$ to be from the family of entropy-regularized $f$-divergences defined as

$$\mathcal{F}_p^f(q) = \int p(\mathbf{x})f(q(\mathbf{x})/p(\mathbf{x}))d\mathbf{x} + \gamma \int q(\mathbf{x})\log q(\mathbf{x})d\mathbf{x}, \quad (4)$$

where $f : \mathbb{R}^+ \to \mathbb{R}$ is a twice-differentiable convex function with $f(1) = 0$. We can understand the first term, the $f$-divergence, as measuring the discrepancy between $q(\mathbf{x})$ and $p(\mathbf{x})$. Popular $f$-divergences include the Kullback-Leibler (KL), Pearson-$\chi^2$ divergence and Jensen-Shannon (JS) divergence. The first term of Eq. 4 thus ensures that the "distance" between $q(\mathbf{x})$ and $p(\mathbf{x})$ decreases along the gradient flow, while the second (differential entropy) term improves expressiveness and prevents the collapse of the gradient flow onto the data points (Ansari et al., 2021).

**From Distributions to Particles.** In this study, we are interested in functional optimization of the form $\min_{q\in\mathcal{P}(\Omega)}\mathcal{F}_p^f(q)$. One way to formulate this optimization is to construct a gradient flow in Wasserstein space. The gradient flow of the functional $\mathcal{F}_q^f(p)$ in Wasserstein space is the curve of measures $\{q_t\}_{t\in\mathbb{R}^+}$ which solves the following Fokker-Planck equation (FPE) (Ambrosio et al., 2005; Villani, 2009):

$$\partial_t q_t(\mathbf{x}) = \text{div}(q_t(\mathbf{x})\nabla_{\mathbf{x}}f'(q_t(\mathbf{x})/p(\mathbf{x})) + \gamma\nabla_{\mathbf{x}}^2 q_t(\mathbf{x}), \quad (5)$$

where div and $\nabla_{\mathbf{x}}^2$ denote the divergence and Lapacian operators respectively, and $f'$ denotes the first derivative of $f$. Eq. 5 is a non-linear partial differential equation, which makes solving for the distribution $q_t$ challenging. Instead of attempting to obtain the WGF at the population density level by solving for $q_t$, we can utilize the connection of FPEs with stochastic differential equations (SDE) and *simulate* the equivalent particle system described by the following SDE:

$$d\mathbf{x}_t = -\nabla_{\mathbf{x}}f'(q_t(\mathbf{x}_t)/p(\mathbf{x}_t))dt + \sqrt{2\gamma}d\mathbf{w}_t, \quad (6)$$

where $d\mathbf{w}_t$ denotes the standard Wiener process. Eq. 6 describes the stochastic evolution of a particle $\mathbf{x}_t$; in image modeling terms, $\mathbf{x}_t$ represents an image sample as it is evolved through the SDE. Eq. 6 and Eq. 5 are equivalent in the sense that the marginal distribution $q_t$ of particles that

evolve under Eq. 6 satisfies the FPE of Eq. 5. In other words, we are able to obtain samples from $q_t$ along the gradient flow of $\mathcal{F}_p^f(q)$ by first drawing samples $\mathbf{x}_0 \sim q_0$ and then simulating the SDE in Eq. 6. Empirically, we simulate the discretized version of Eq. 6 using the Euler-Maruyama method

$$\mathbf{x}_{k+1} = \mathbf{x}_k - \eta \nabla_{\mathbf{x}} f'(q_k(\mathbf{x}_k)/p(\mathbf{x}_k)) + \sqrt{2\gamma\eta}\boldsymbol{\xi}_k, \tag{7}$$

where $\boldsymbol{\xi}_k \sim \mathcal{N}(0, I)$, $\eta$ is the step size and the time interval $[0, K]$ is discretized into equal intervals.

## 3.1 GRADIENT FLOW VIA DENSITY RATIO ESTIMATION

Simulating the gradient flow in Eq. 7 requires an estimate of the density ratio $q_t(\mathbf{x})/p(\mathbf{x})$, which is unavailable to us. We would like to leverage the Bregman divergence to train an estimator $r_\theta(\mathbf{x})$, such that we can simulate the gradient flow numerically as

$$\mathbf{x}_{k+1} = \mathbf{x}_k - \eta \nabla_{\mathbf{x}} f'(r_\theta(\mathbf{x}_k)) + \nu\boldsymbol{\xi}_k, \tag{8}$$

where we combine the constants of the Gaussian noise term into a single hyperparameter $\nu$. However, training such an estimator requires access to samples $\mathbf{x}_t \sim q_t(\mathbf{x})$. Unlike diffusion models (Ho et al., 2020; Song et al., 2020), where the time-dependent ground truths $\mathbf{x}_t \sim q(\mathbf{x}_t|\mathbf{x}_0)$ can be obtained analytically, we do not have access to the ground truth samples along the gradient flow. In early experiments, we attempted to draw samples $\mathbf{x}_k$ along every step of the flow when simulating Eq. 8 during training. However, this resulted in poor performance that regularly diverges as the $\mathbf{x}_{1:K}$ drawn changes at every iteration as parameters $\theta$ are updated, resulting in the lack of a stable learning signal that fixed ground truths would provide.

Instead of training the model on $\mathbf{x}_{1:K}$, we propose to only draw samples $\mathbf{x}_K$ by simulating Eq. 8 for the full $K$ steps. Consider a density ratio estimator with parameters $\theta_t$ at training iteration $t$. The $\mathbf{x}_K$ are drawn from the distribution $\tilde{q}_t(\mathbf{x}_K)$ given by

$$\tilde{q}_t(\mathbf{x}_K) = \int q_0(\mathbf{x}) M_{\theta_t}(\mathbf{x}_K|\mathbf{x}) d\mathbf{x} \tag{9}$$

where $M_\theta(\mathbf{x}_K|\mathbf{x})$ is the transition kernel of simulating Eq. 8 using $r_{\theta_t}(\mathbf{x})$. We optimize $r_{\theta_t}(\mathbf{x})$ for the $t$ training iteration using the Bregman divergence

$$\mathcal{L}(\theta_t) = \widehat{\mathbb{E}}_p[\partial g(r_{\theta_t}(\mathbf{x}))r(\mathbf{x}) - g(r_{\theta_t}(\mathbf{x}))] - \widehat{\mathbb{E}}_{\tilde{q}_t}[\partial g(r_{\theta_t}(\mathbf{x}))] \tag{10}$$

where expectation over $p(\mathbf{x})$ can be estimated using samples drawn from the dataset. Similar to Ansari et al. (2021), we are effectively approximating the time-dependent density ratio with a stale estimate $r_\theta(\mathbf{x}) = \tilde{q}(\mathbf{x})/p(\mathbf{x})$. As training progresses, the samples $\mathbf{x}_K \sim \tilde{q}_t(\mathbf{x}_K)$ improve correspondingly with $r_{\theta_t}(\mathbf{x})$. We hypothesize that training $r_\theta(\mathbf{x})$ on samples of improving quality allows it to learn the density ratio across the gradient flow *without* the need for explicit time-dependence. We validate our hypothesis experimentally in Sec. 5.2, where we show that despite a potentially time-independent formulation, our model does not collapse to a single density ratio estimate and has implicitly learned the density ratio over the flow. We find this result noteworthy and motivates further investigations into the necessity of explicit time embedding in relevant frameworks such as diffusion models.

Our training scheme bears resemblance to Short-Run EBM (Nijkamp et al., 2019), where the model draws samples from an approximation of the true Langevin dynamics. At each training step, DGGF learns to refine its estimate of the density ratio $r_\theta(\mathbf{x})$ by looking at positive samples from the data distribution and negative samples drawn from its implicit distribution. However, despite the similarity, DGGF is fundamentally distinct from EBMs: DGGF is formulated as the path of steepest descent in Wasserstein space, while EBMs are derived from maximum likelihood. We further elaborate on the salient distinctions from EBMs and other models in Sec. 4. Once DGGF is trained, sampling at test time is simply a matter of running Eq. 8 directly in data space. We provide training and sampling pseudocode in Algorithms 1 and 2 respectively.

**Choices for $f$-divergences and Bregman divergence.** DGGF allows for flexibility in the choice of $f$-divergence in the gradient flow, as well as the choice of $g$ in the Bregman divergence objective. We consolidate a list of common $f$-divergences and their first derivatives that we study in this paper in Table 2 in the appendix.

Certain forms for $g$ pair naturally with specific $f$-divergences for the gradient flow, as they simplify the calculations needed. In our experiments, we utilize two forms of $g$: the first is when $g(t) = \frac{1}{2}(t-1)^2$, which corresponds to the Least-Squares Importance Fitting (LSIF) objective:

$$\mathcal{L}_{LSIF}(\theta) = \frac{1}{2}\widehat{\mathbb{E}}_p[r_\theta(\mathbf{x})]^2 - \widehat{\mathbb{E}}_{q_t}[r_\theta(\mathbf{x})], \tag{11}$$

and the second is $g(t) = t \log t - (1+t)\log(1+t)$, which corresponds to the Logistic Regression (LR) objective:

$$\mathcal{L}_{LR}(\theta) = -\widehat{\mathbb{E}}_p\left[\log \frac{1}{1+r_\theta(\mathbf{x})}\right] - \widehat{\mathbb{E}}_{q_t}\left[\log \frac{r_\theta(\mathbf{x})}{1+r_\theta(\mathbf{x})}\right]. \tag{12}$$

Due to numerical compatibility issues that we discuss in Appendix B, we pair the LSIF objective with the Pearson-$\chi^2$ divergence and the LR objective with the KL, JS and logD divergences. We abbreviate them as LSIF-$\chi^2$, LR-KL, LR-JS and LR-logD respectively and study these pairings in our experiments in Sec. 5.

## 3.2 DENSITY CHASM AND DATA-DEPENDENT PRIORS

In principle, the gradient flow formulation does not place any restrictions on the prior distribution $q_0(\mathbf{x})$. In the $t \to \infty$ limit, the gradient flow is guaranteed to converge to $p(\mathbf{x})$ (see Appendix D for a proof). However, empirically, we observe that when implementing the flow with a finite number of steps, the choice of prior distribution can significantly affect model performance.

This observation may be attributed to the density chasm problem (Rhodes et al., 2020). Consider two densities $q(\mathbf{x})$ and $p(\mathbf{x})$ which differ greatly, for example as measured by their KL divergence. A binary classifier that is trained on distinguishing the two distributions, which is equivalent to density ratio estimation using the "density ratio trick", can obtain near perfect accuracy while learning a relatively poor estimate of the density ratio (Ansari et al., 2021). A simple example would be two narrow Gaussians with means that are far apart. A classifier can learn a trivial boundary with near perfect accuracy, such as a straight line between the two modes, without having to estimate the density ratio between the two distributions accurately.

We found that using common priors for $q_0(\mathbf{x})$ such as a uniform distribution led to poorer sample quality due to the large chasm between the simple distribution and the complex multimodal data distribution. Inspired by generation from seed distributions with robust classifiers (Santurkar et al., 2019), we leverage a *data-dependent* prior by fitting a simple multivariate Gaussian to the training dataset, and sampling $\mathbf{x}_0$ from the data-dependent prior

$$q_0(\mathbf{x}) = \mathcal{N}(\boldsymbol{\mu}_D, \boldsymbol{\Sigma}_D), \quad \text{where } \boldsymbol{\mu}_D = \mathbb{E}_D[\mathbf{x}], \quad \boldsymbol{\Sigma}_D = \mathbb{E}_D[(\mathbf{x} - \boldsymbol{\mu}_D)^T(\mathbf{x} - \boldsymbol{\mu}_D)] \tag{13}$$

where $D$ represents the training dataset. Samples from this prior are poor but this approach is sufficient to cross the density chasm; we visualize some samples from the prior in Fig. 8, and experimental results support this approach (Sec. 5.2).

---

**Algorithm 1** Training

**repeat**
  Sample $\mathbf{x}_p \sim p(x), \mathbf{x}_0 \sim q_0(\mathbf{x})$
  **for** $j \leftarrow 1, K$ **do**
    Obtain $x_K$ from $x_0$ by simulating Eq. 8.
  **end for**
  Update $\theta$ according to

$$\nabla_\theta[g'(r_\theta(\mathbf{x}_p))r(\mathbf{x}_p) - g(r_\theta(\mathbf{x}_p)) - g'(r_\theta(\mathbf{x}_K))]$$

**until** converged

**Algorithm 2** Sampling

  Sample $\mathbf{x}_0 \sim q_0(\mathbf{x})$
  **for** $j \leftarrow 1, K$ **do**
    Obtain $x_K$ from $x_0$ by simulating Eq. 8.
  **end for**
  **return** $\mathbf{x}_K$

---

## 4 RELATED WORKS

Gradient flows are a general framework for constructing the steepest descent curve of a given functional, and consequently have been used in optimizing a variety of distance metrics, ranging from

the $f$-divergence (Gao et al., 2019; 2022; Ansari et al., 2021; Fan et al., 2021), maximum mean discrepancy (Arbel et al., 2019; Mroueh & Nguyen, 2021), Sobolev distance (Mroueh et al., 2019) and related forms of the Wasserstein distance (Liutkus et al., 2019). Recent works have also explored the interpretation of the FPE as a continuous normalizing flow for generative modeling (Xu et al., 2022). A well-known method to simulate WGFs at the population density level is using the Jordan, Kinderlehrer, and Otto (JKO) scheme (Jordan et al., 1998), which approximates the dynamics of the FPE through an iterative time-discretization update. The JKO scheme requires computation of the 2-Wasserstein distance and a free energy functional, which are generally intractable. Several works have been proposed to circumvent this problem: Mokrov et al. (2021) leverages Brenier's theorem and convex neural networks to approximately estimate $\mathcal{W}_2$, while Fan et al. (2021) leverages the Fenchel conjugate to evaluate the free energy as $f$-divergences. Our work avoids the JKO scheme entirely by adopting a particle-based approach, where we use the Euler-Maruyama method to simulate the gradient flow, parameterized by neural networks. In this way, we avoid the need to estimate $\mathcal{W}_2$, as well as the dual optimization needed in the variational formulation of Fan et al. (2021).

More closely related to DGGF are other particle-based approaches. Liutkus et al. (2019) uses a non-parametric approach to optimize the Sliced Wasserstein distance, where the marginal distribution of the particles is computed empirically to approximate $q_t(\mathbf{x})$. DG$f$low proposes to leverage pretrained GAN discriminators as density ratio estimators for sample refinement. DGGF can be seen as a general case of DG$f$low, where we recover DG$f$low if we fix the prior $q_0$ to be the implicit distribution defined by the GAN generator. When $q_0$ is chosen as a simple prior, DGGF can perform generation from scratch, as opposed to purely refinement. Similar to DGGF, VGrow and EPT train deep density ratio estimators using the Bregman divergence and apply them to the task of unconditional image generation. However, there are several key differences that distinguish DGGF from VGrow and EPT. As VGrow is a specific instance of EPT with the logistic regression objective, we focus on EPT while noting that our discussion applies to both methods. EPT utilizes a formalism that performs generation on a predetermined batch of particles during the training process. This causes the estimator to converge to a single density ratio estimate after training, preventing further sampling at test time due to zero gradients. In the same vein, EPT also has poor generative performance when formulated in the data space[1]. It was necessary to amortize sample generation to an auxiliary generator and formulate the WGF in latent space $z$. By flowing $\mathbf{z}$ and matching $G_\theta(\mathbf{z}) = \mathbf{x}$, the generator can be sampled at test time. We show from our results in Sec. 5 that generation can be done entirely in the data space, without the complexity of training an additional generator.

Our model also shares similarities with EBMs (LeCun et al., 2006), which model the data likelihood using a Boltzmann distribution. The model can be sampled efficiently with Langevin dynamics (Welling & Teh, 2011), which requires an infinitely long chain for proper mixing. Similar to DGGF, Nijkamp et al. (2019) reinitializes the chain at every training iteration, effectively sampling from an approximation of the true Langevin dynamics. Fundamentally, the key difference is DGGF is formulated by finding the steepest descent curve in Wasserstein space, while EBMs are derived from maximum likelihood estimation. As shown in prior works (Jordan et al., 1998; Liu et al., 2019), Langevin dynamics can be derived from WGFs when $f$ is chosen as the KL divergence, which we reproduce in Appendix E. The density ratio formulation of DGGF allows us the flexibility of composition with other density ratio estimators, as well as choosing the source and target distributions of the flow, which permits applications such as composition with pretrained classifiers (Sec. 5.3) and image-to-image translation (Sec. 5.4). These are not directly applicable with EBMs, and adaptations to such domains require significant theoretical and experimental modifications to the generative framework (Zhao & Chen, 2021; Sasaki et al., 2021).

## 5 EXPERIMENTS

In this section, we present empirical results of the generative performance of DGGF on various common image datasets across different resolutions. Our goal is to demonstrate that DGGF is able to generate diverse, high-quality images across various $f$-divergences and Bregman divergence objectives, as well as generalizes to other tasks such as class-conditional generation and image-to-image

---

[1] Based on experiments of EPT in Sec. 5 and on unpublished scores cited by the authors; please see `https://openreview.net/forum?id=awMgJJ9H-0q`.

[1] Score as reported in Gao et al. (2020).

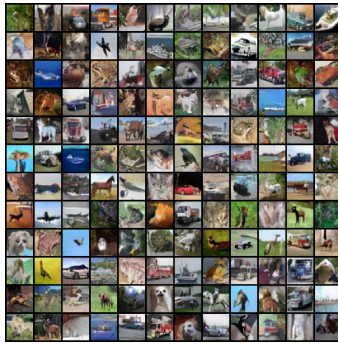 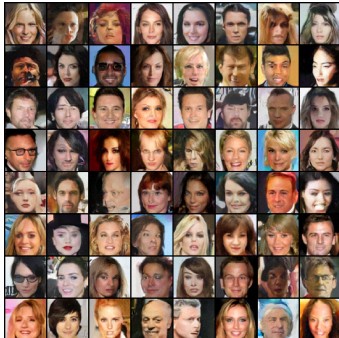 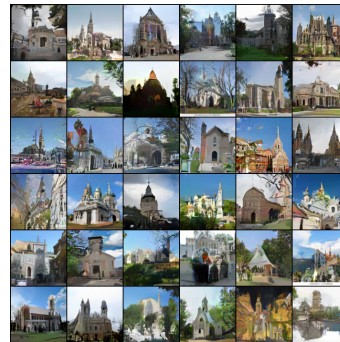

Figure 2: Samples from DGGF-DDP on CIFAR10 $32^2$, CelebA $64^2$ and LSUN Church $128^2$ using LSIF-$\chi^2$. More results using various BD objectives and $f$-divergences can be found in Appendix.

translation. We will use DGGF-DDP to abbreviate experiments with the data-dependent prior and DGGF-UP for ablation experiments with the uniform prior.

## 5.1 SETUP

We test DGGF with unconditional generation on CIFAR10, CelebA and LSUN Church datasets, class-conditional generation on CIFAR10 and image-to-image translation on the Cat2dog dataset. All pixel values are normalized to the range [-1, 1]. For CIFAR10, we keep the resolution of the images at $32 \times 32$, while for CelebA and LSUN Church we resize them to $64 \times 64$ and $128 \times 128$, respectively. We use modified ResNet architectures for all experiments in this study. See Appendix G for more details.

## 5.2 IMAGE GENERATION

In Fig. 2, we show uncurated samples of DGGF-DDP on different combinations of $g$ and $f$-divergences. Visually, our model is able to produce high-quality samples on a variety of datasets up to resolutions of $128 \times 128$, surpassing existing gradient flow techniques (Gao et al., 2019; 2022). More samples with other $f$-divergences can be found in Appendix I. In Table 1, we show the FID scores of DGGF-DDP in comparison with relevant baselines utilizing different generative approaches. On CIFAR10, our model performs com-

Table 1: CIFAR10 and CelebA scores.

| Model | FID ↓ |
|---|---|
| **CIFAR10 32 × 32** | |
| EPT (no outer loop) (Gao et al., 2022) | 46.63 |
| JKO-Flow (Fan et al., 2021) | 23.7 |
| IGEBM (Du & Mordatch, 2019) | 40.58 |
| SNGAN (Miyato et al., 2018) | 21.7 |
| PixelCNN (Van Oord et al., 2016) | 65.93 |
| NVAE (Vahdat & Kautz, 2020) | 51.67 |
| NCSN (Song & Ermon, 2019) | 25.32 |
| DGGF-DDP (LSIF-$\chi^2$) | 28.12 |
| DGGF-DDP (LR-KL) | 28.80 |
| DGGF-DDP (LR-JS) | 29.92 |
| DGGF-DDP (LR-logD) | 30.72 |
| DGGF-UP (LSIF-$\chi^2$) | 35.11 |
| DGGF-UP (LR-KL) | 39.90 |
| **CelebA 64 × 64** | |
| NCSN (Song & Ermon, 2019) | 26.89 |
| NVAE (Vahdat & Kautz, 2020) | 14.74 |
| EBM-SR (Nijkamp et al., 2019) | 23.02[2] |
| DGGF-DDP (LSIF-$\chi^2$) | 22.42 |
| DGGF-DDP (LR-KL) | 22.88 |

parably with SNGAN and the score-based NCSN, while outperforming the baseline EBM, autoregressive method PixelCNN. Our method strongly outperforms EPT without an auxiliary generator (no outer loop), which is the gradient flow baseline utilizing a density ratio method. In comparison to the WGF baselines, DGGF strongly outperforms EPT without an auxiliary generator (no outer loop), while performing comparably with JKO-Flow. For CelebA, our model is outperformed by the state-of-the-art variational autoencoder NVAE, but outperforms NCSN and Short-Run EBM.

To provide intuition for the gradient flow process, we provide intermediate samples for the LSUN Church dataset in Fig. 5 of the appendix, which visualizes how samples drawn from the data-dependent prior is evolved to a high quality sample. In this scenario, the prior contains low level features such as the color of the sky and rough silhouette of the building. The gradient flow retains these coarse features, but generates the higher frequency details necessary to create a realistic im-

age. This matches the intuition that the gradient flow is the steepest descent curve, as the shortest path would mean changing as little of the image as possible to reach the target distribution. We also visualize samples obtained by interpolating in the prior space for CelebA in Fig. 6 of the appendix. Despite the use of a relatively complex prior, the model is able to smoothly interpolate in the latent space, indicating the model has learnt a semantically relevant latent representation that is characteristic of a valid generative model. Finally, to verify that our model has not merely memorized the dataset, particularly due to concerns that the prior is fitted from data, we show nearest neighbor samples of the generated images in the training set in Appendix H. We can clearly see that samples produced by DGGF-DDP are distinct from the closest samples in the training set, which tells us that DGGF-DDP is capable of generating new and diverse samples beyond the data it was trained on.

**Model Estimate of KL over Flow.** As our model outputs the density ratio $r_\theta(\mathbf{x}_k) = q(\mathbf{x}_k)/p(\mathbf{x}_k)$ throughout the gradient flow, our model can be interpreted as estimating the KL divergence $D_{KL}(q(\mathbf{x}_k)||p(\mathbf{x}_k)) = \int q(\mathbf{x}_k) \log(q/p)(\mathbf{x}_k)d\mathbf{x}_k = \mathbb{E}_{\mathbf{x}_k}[\log r_\theta(\mathbf{x}_k)]$ where the expectation is taken over the batch of samples being evolved. We denote this estimate as $\widehat{D}_{KL}$. We show that this estimate is valid in Fig. 3, which plots $\widehat{D}_{KL}$ over the gradient flow for both DDP and UP (the UP results serve as ablations in the next paragraph). Focusing on the DDP results, we observe that $\widehat{D}_{KL}$ decreases monotonically over the flow, which agrees with the notion that $q(\mathbf{x}_k)$ approaches $p(\mathbf{x}_k)$, as seen in Fig. 5. This validates our hypothesis that despite the use of a stale estimate, the model did not collapse to a single density ratio and has learnt a valid density ratio over the flow. As evident from Fig. 5, the sample quality improves progressively over the flow as $\widehat{D}_{KL}$ decreases. As such, this provides DGGF with an interpretable diagnostic of its own sample quality that comes innately with the density ratio formulation.

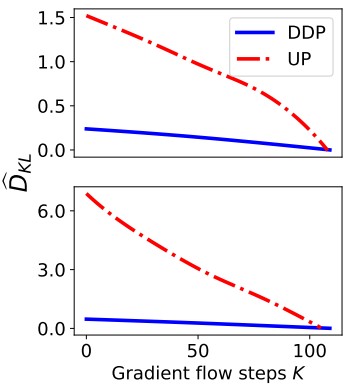

Figure 3: KL estimate $\widehat{D}_{KL}$ of the gradient flow for CIFAR10 generation for both DDP and UP for LSIF-$\chi^2$ (upper) and LR-KL (lower).

**Ablations with Uniform Prior.** We motivate the use of the data-dependent prior by conducting ablation experiments with $q_0$ being a uniform prior, $\mathbf{x}_0 \sim U[-1, 1]$. All hyperparameters are kept identical to DDP experiments to isolate the effects of the choice of prior distribution. We include qualitative samples of DGGF-UP in Fig. 18 in the appendix. The quantitative results can be seen from the FID scores in Table 1. Visually, DGGF-UP produces diverse and appealing samples even with a uniform prior. However, when comparing quantitative scores we observe that the use of DDP improves results significantly. Support for the density chasm hypothesis can be found by comparing the $\widehat{D}_{KL}$ of DDP and UP in Fig. 3. For both LSIF-$\chi^2$ and LR-KL, DGGF-UP has a significantly larger $\widehat{D}_{KL}$ at the start of the flow as compared to DGGF-DDP. This corresponds to our intuition, as the uniform prior is 'farther' from the data distribution as compared to the data-dependent prior. As a result, the model has to push the particles over a larger density chasm, leading to poorer performance given a fixed computational budget.

## 5.3 CONDITIONAL GENERATION WITH ROBUST CLASSIFIERS

The density ratio framework allows us to compose different density ratio estimators together, therefore allowing us to construct WGFs between distributions different from those in training. To illustrate this, consider a multiclass classifier which predicts the probability that a given image belongs to any one of $N$ classes. We show in Appendix F that we can express such classifiers as a density ratio $p(y = n|\mathbf{x}) = N^{-1}p(\mathbf{x}|y = n)/p(\mathbf{x})$. We can thus obtain a conditional density ratio estimator $r_\theta(\mathbf{x}_t|y = n) = q_t(\mathbf{x}_t)/p(\mathbf{x}_t|y = n)$ by composing our unconditional estimator $r_\theta(\mathbf{x}_t)$ with the classifier output (see Appendix F):

$$r_\theta(\mathbf{x}_t|y = n) = \frac{1}{N}r_\theta(\mathbf{x}_t)p(y = n|\mathbf{x}_t)^{-1}. \tag{14}$$

When $r_\theta(\mathbf{x}_t|y = n)$ is used in simulating the WGF in Eq. 8, we obtain a class-conditional generative model. This is conceptually similar to the idea proposed in Song et al. (2020), where an unconditional score model $\nabla_\mathbf{x} \log p_t(\mathbf{x}(t))$ is composed with a time-dependent classifier $\nabla_\mathbf{x} \log p_t(y|\mathbf{x}(t))$

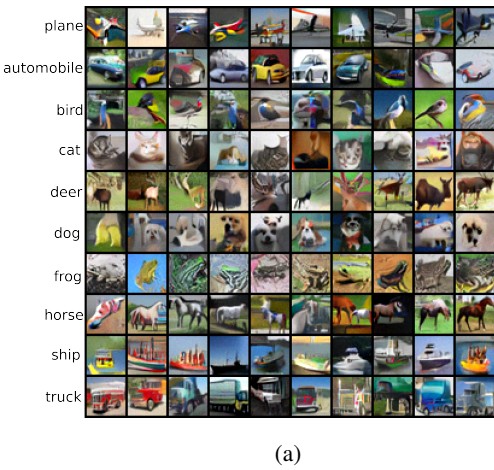 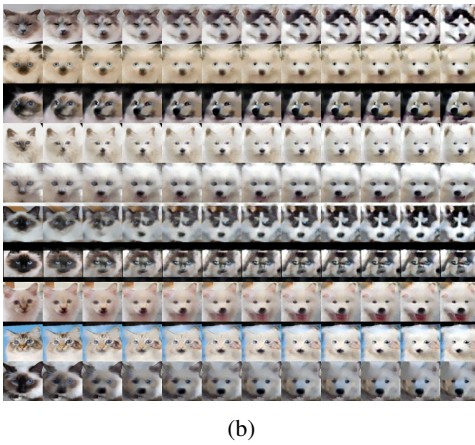

(a)                                                    (b)

Figure 4: (a) Class-conditional samples from composition with a robust classifier. (b) Image-to-image translation process from cat to dog images using DGGF.

to form a class-conditional model. However, whereas Song et al. (2020) requires the separate training of a time-dependent classifier, our formulation allows us to use off-the-shelf pretrained classifiers with *no further retraining*. Inspired by earlier work on image synthesis with robust classifiers (Santurkar et al., 2019), we found that using a pretrained adversarially-robust classifier was necessary in obtaining useful gradients for the gradient flow. We show our results in Fig. 4a, where each row represents conditional samples of each class in the CIFAR10 dataset.

## 5.4 Unpaired Image-to-image Translation

Our framework can also be directly applied to unpaired image-to-image-translation (I2I). We simply fix the prior distribution $q_0(\mathbf{x})$ to a source domain and $p(\mathbf{x})$ to a target domain. We then train the model in exactly the same manner as unconditional generation (see Algorithm 1).

We test our I2I model on the Cat2dog dataset (Lee et al., 2018). From Fig. 4b, DGGF is able to smoothly translate images of cats to dogs while maintaining relevant semantic features of the image. For example, the background colors of the image and the pose of the cat are unchanged—a cat that is facing a certain direction is translated to a dog that faces the same direction. We also observe that the facial tones of the cat is preserved—a cat with light fur is translated to a dog with light fur. CycleGAN (Zhu et al., 2017) achieves better FID scores than DGGF (Table 4 in the appendix) but like many I2I methods (Lee et al., 2018; Choi et al., 2020; Zhao & Chen, 2021; Nie et al., 2021), CycleGAN incorporates specific inductive biases, such as dual generators and discriminators together with the cycle-consistency loss. Incorporating such inductive biases into the gradient flow process can improve the translation and would make for interesting future work.

## 6 Conclusion

In this paper, we proposed DGGF, a method to simulate the Wasserstein gradient flow between two distributions that minimizes the entropy-regularized $f$-divergence. As constructing such a flow requires an estimate of the density ratio, we showed how to leverage the Bregman divergence to train a deep density ratio estimator that is able to synthesize diverse images of high quality. We showed that the modularity of DGGF allows for composition with external density ratio estimators, as well as direct application to tasks such as unpaired image-to-image translation. Given the flexibility of choosing the source and target distribution, future work can investigate different choices of the two distributions that could correspond to entirely new applications. Another possible avenue is investigating how we can incorporate task-specific inductive biases into the gradient flow process, which should allow the WGF to perform better on the problem at hand.

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

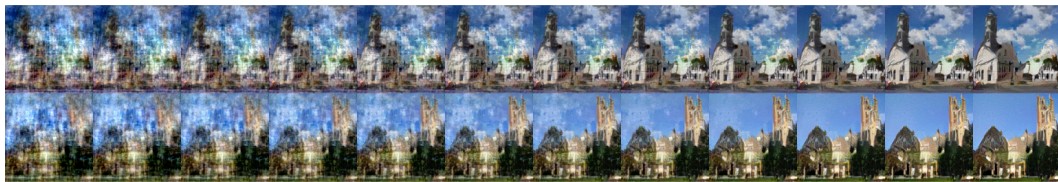

Figure 5: Illustration of the gradient flow process for LSUN Church, starting from a sample from the data-dependent prior on the leftmost column.

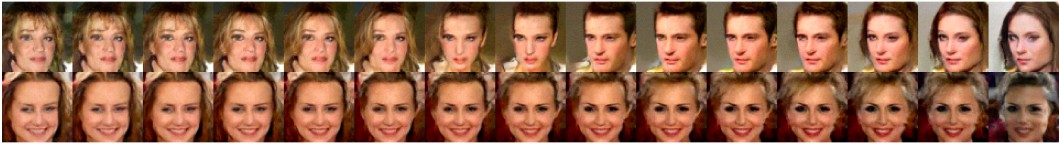

Figure 6: Interpolation results between leftmost and rightmost samples with CelebA.

## A    TOY DATASETS

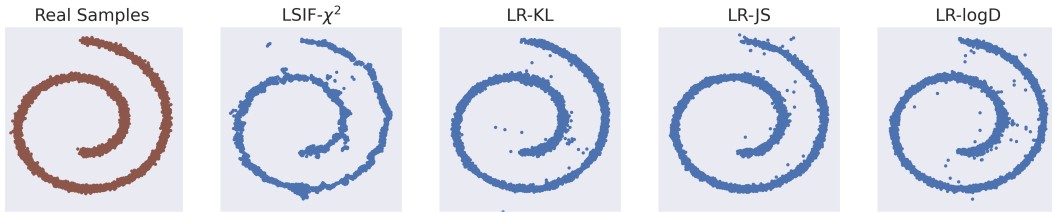

Figure 7: Comparison of different DGGF pairings of Bregman and $f$-divergences on the 2DSwiss-roll dataset.

To affirm that samples generated by DGGF indeed converge to the target distribution $p(\mathbf{x})$, we train DGGF on the synthetic 2DSwissroll dataset. The density ratio estimator is parameterized by a simple feedforward multilayer perceptron. We train the model to flow samples from the prior $q_0(\mathbf{x}) = \mathcal{N}(0, \mathbf{I})$ to the target distribution, which we sample from the `make_swiss_roll` function in `scikit-learn`. We plot the results in Fig. 7, from which we can see that the model indeed converges to $p(\mathbf{x})$ successfully for all combinations of $f$ and $g$.

## B    BREGMAN DIVERGENCE AND $f$-DIVERGENCE PAIRING

When computing the LR objective Eq. 12, we find that we run into numerical stability issues when letting $r_\theta(\mathbf{x})$ be the output of an unconstrained neural network and subsequently taking the required logarithms in Eq. 12. To circumvent this issue, we let $r_\theta(\mathbf{x})$ be expressed as the exponential of the neural network's output, i.e., the output of the neural network is $\log r_\theta(\mathbf{x})$. This formulation naturally lends itself to the gradient flow of the KL, JS and logD divergences, whose first derivatives $f'$ that is required in Eq. 8 are also logarithmic functions of $r_\theta(\mathbf{x})$, as seen from Table. 2. We can thus utilize numerically stable routines in existing deep learning frameworks, avoiding the need for potentially unstable operations like exponentiations (see Appendix C for details). As such, we pair LR with the aforementioned divergences and abbreviate the combinations as LR-KL, LR-JS, LR-logD. We did not run into such stability issues for the LSIF objective (Eq. 11) as the model learns to automatically output a non-negative scalar over the course of training, hence for LSIF we allow the neural network to estimate $r_\theta(x)$ directly and pair it with the Pearson-$\chi^2$ divergence. We abbreviate this pairing as LSIF-$\chi^2$.

## C  STABLE COMPUTATION OF LR AND $f$-DIVERGENCES

As mentioned in Sec. B, computing the logarithm of unconstrained neural networks leads to instabilities in the training process. This is a problem when computing the LR objective in Eq. 12 and the various first derivatives of $f$-divergences. We can circumvent this problem by letting $r_\theta(\mathbf{x})$ be expressed as the exponential of the neural network and use existing stable numerical routines to avoid intermediate computations that lead to the instabilities (for example, computing logarithms and exponentials directly). Let us express the neural network output as $NN_\theta(x) \triangleq \log r_\theta(x)$. The LR objective can then be rewritten as

$$\mathcal{L}_{LR}(\theta) = -\widehat{\mathbb{E}}_p \left[\log \frac{1}{1+r_\theta(\mathbf{x})}\right] - \widehat{\mathbb{E}}_{q_t} \left[\log \frac{r_\theta(\mathbf{x})}{1+r_\theta(x)}\right] \tag{15}$$

$$= -\widehat{\mathbb{E}}_p \left[\texttt{logsigmoid}(-NN_\theta(x))\right] - \widehat{\mathbb{E}}_{q_t} \left[\texttt{logsigmoid}(NN_\theta(x))\right] \tag{16}$$

where $\texttt{logsigmoid}(x) = \log \frac{1}{1+\exp(-x)}$, which has stable implementations in modern deep learning libraries.

Similarly for the $f$-divergences whose first derivatives involve logarithms, we can calculate them stably as

$$f'_{KL}(r(\mathbf{x})) = \log r(\mathbf{x}) + 1 = NN_\theta(\mathbf{x}) + 1 \tag{17}$$

$$f'_{JS}(r(\mathbf{x})) = \log \frac{2r(\mathbf{x})}{1+r(\mathbf{x})} = \log 2 + \texttt{logsigmoid}(NN_\theta(\mathbf{x})) \tag{18}$$

$$f'_{logD}(r(\mathbf{x})) = \log(r+1) + 1 = -\texttt{logsigmoid}(-NN_\theta(\mathbf{x})) + 1. \tag{19}$$

## D  PROOF OF CONVERGENCE OF GRADIENT FLOW

We provide a simple proof of the convergence of the Wasserstein gradient flow in the $t \to \infty$ limit.

**Theorem 1.** *Let the functional $\mathcal{F}_p^f(q^t)$ be defined as*

$$\mathcal{F}_p^f(q^t) = \int p(\mathbf{x}) f(q^t(\mathbf{x})/p(\mathbf{x})) d\mathbf{x} + \gamma \int q^t(\mathbf{x}) \log q^t(\mathbf{x}) d\mathbf{x}. \tag{20}$$

*$F_p^f(q^t)$ is non-increasing as a function of time and converges to the global minimum in the $t \to \infty$ limit.*

*Proof.*

$$\frac{\partial \mathcal{F}_p^f(q^t)}{\partial t} = \langle \nabla_{W_2} \mathcal{F}_p^f(q^t), \frac{\partial q^t}{\partial t} \rangle \tag{21}$$

$$= -||\nabla_{W_2} \mathcal{F}_p^f(q^t)||^2. \tag{22}$$

where in the first line we apply the chain rule, and in the second line we use the definition of the gradient flow in the Wasserstein space, $\frac{\partial q^t}{\partial t} = -\nabla_{W_2} \mathcal{F}_p^f(q^t)$ (analogous to the gradient flow in Euclidean space of Eq. 1). $\qquad\square$

## E  CONNECTIONS WITH LANGEVIN DYNAMICS

In this section we demonstrate the connection between WGFs and Langevin dynamics, which have also been studied in prior works such as Jordan et al. (1998); Liu et al. (2019). Langevin dynamics is a MCMC method that is able to produce samples from a probability density $p(\mathbf{x})$ using only the score function $\nabla_\mathbf{x} \log p(\mathbf{x})$. Let $\epsilon$ be the step size and $\mathbf{x}_0 \sim \pi(\mathbf{x})$ be an initial sample drawn from a prior distribution, Langevin dynamics iteratively updates $\mathbf{x}_k$ as

$$\mathbf{x}_{k+1} = \mathbf{x}_k + \frac{\epsilon}{2} \nabla_\mathbf{x} \log p(\mathbf{x}_k) + \sqrt{\epsilon} \boldsymbol{\xi}_k, \tag{23}$$

Table 2: $f$-divergences and their first derivatives $f'$.

| $f$-divergence | $f$ | $f'$ |
|---|---|---|
| Pearson-$\chi^2$ | $(r-1)^2$ | $2(r-1)$ |
| KL | $r \log r$ | $\log r + 1$ |
| JS | $r \log r - (r+1) \log \frac{r+1}{2}$ | $\log \frac{2r}{r+1}$ |
| log D | $(r+1) \log(r+1) - 2 \log 2$ | $\log(r+1) + 1$ |

where $\xi_k \sim \mathcal{N}(0, I)$. As shown by Welling & Teh (2011), as $k \to \infty$ and $\eta \to 0$, then $\mathbf{x}_k \sim p(\mathbf{x})$ under certain regularity conditions. Langevin dynamics are most notably utilized as a method to sample from EBMs, which model $p(\mathbf{x})$ as a Boltzmann distribution $p_\theta(\mathbf{x}) = \exp(-E_\theta(\mathbf{x}))/Z(\theta)$.

We show the connection to WGFs by considering the discretized SDE Eq. 7 with $f' = \log r + 1$ corresponding to the KL divergence:

$$\mathbf{x}_{k+1} = \mathbf{x}_k - \eta \nabla_\mathbf{x} \log(q_k(\mathbf{x}_k)/p(\mathbf{x}_k)) + \sqrt{2\gamma\eta}\xi_k. \tag{24}$$

Upon decomposing the logarithm term, we immediately see that the WGF gives us

$$\mathbf{x}_{k+1} = \mathbf{x}_k - \underbrace{\eta \nabla_\mathbf{x} \log q_k(\mathbf{x}_k)}_{\text{prior downweighting}} + \underbrace{\eta \nabla_\mathbf{x} \log p(\mathbf{x}_k) + \sqrt{2\gamma\eta}\xi_k}_{\text{Langevin dynamics}}. \tag{25}$$

We exactly recover Langevin dynamics in the last two terms if we set the hyperparameters $\eta = \epsilon/2$ and $\gamma = 1$. The emergence of the term we call 'prior downweighting' can be interpreted as pushing the samples away from $q(\mathbf{x})$ by reducing its log-likelihood. Hence, WGF with $f$ corresponding to the KL divergence can be intuitively understood as an 'enhanced' version of Langevin dynamics, where samples are pushed in the direction which not only increases $\log p(\mathbf{x})$, but also explicitly decreases $\log q(\mathbf{x})$.

## F  CLASSIFIERS ARE DENSITY RATIO ESTIMATORS

To perform class-conditional generation in the DGGF framework, we would like to estimate the density ratio of a certain class over the data distribution: $p(\mathbf{x}|y = n)/p(\mathbf{x})$. With Bayes rule, we can write this as

$$\frac{p(\mathbf{x}|y = n)}{p(\mathbf{x})} = \frac{p(y = n|\mathbf{x})p(\mathbf{x})/p(y = n)}{p(\mathbf{x})} \tag{26}$$

$$= \frac{p(y = n|\mathbf{x})}{p(y = n)}. \tag{27}$$

The denominator term $p(y = n)$ can be viewed as a constant, e.g., assume the $N$ classes are equally distributed, then $p(y = n) = 1/N$. Therefore, we have that the class probability given by the softmax output of a classifier is actually a density ratio:

$$Np(y = n|\mathbf{x}) = \frac{p(\mathbf{x}|y = n)}{p(\mathbf{x})}. \tag{28}$$

We can use this equation to convert an unconditional DGGF to a class-conditional generator. Recall the gradient flow equation:

$$d\mathbf{x}_t = -\nabla_\mathbf{x} f'(r_\theta(\mathbf{x}_t))dt + \sqrt{2\gamma}d\mathbf{w}_t \tag{29}$$

We can multiply the inverse of the classifier output with $r_\theta(\mathbf{x}_t) = q_t(\mathbf{x}_t)/p(\mathbf{x}_t)$ to get a density ratio between $q_t(\mathbf{x}_t)$ and the conditional data distribution $p(\mathbf{x}_t|y = n)$:

$$r_\theta(\mathbf{x}_t)p(y = n|\mathbf{x}_t)^{-1} = \frac{q_t(\mathbf{x}_t)}{p(\mathbf{x}_t)} \frac{Np(\mathbf{x}_t)}{p(\mathbf{x}_t|y = n)} = N\frac{q_t(\mathbf{x}_t)}{p(\mathbf{x}_t|y = n)}. \tag{30}$$

That is, we took our unconditional model and converted it to a conditional generative model by composing it with a pretrained classifier. To get the correct class-conditional density ratio, we should therefore compute

$$r_\theta(\mathbf{x}_t|y = n) = \frac{1}{N}r_\theta(\mathbf{x}_t)p(y = n|\mathbf{x}_t)^{-1} \tag{31}$$

and use this conditional density ratio estimator in the WGF SDE.

Table 3: Network structures for the density ratio estimator $r_\theta(\mathbf{x})$.

| CIFAR10 | CelebA 64 | LSUN Church 128 | Cat2dog 64 |
|---|---|---|---|
| 3×3 Conv2d, 128 | 3×3 Conv2d, 64 | 3×3 Conv2d, 64 | 3×3 Conv2d, 64 |
| 3 × ResBlock 128 | ResBlock Down 64 | ResBlock Down 64 | ResBlock Down 64 |
| ResBlock Down 256 | ResBlock Down 128 | ResBlock Down 128 | ResBlock Down 128 |
| 2 × ResBlock 256 | ResBlock 128 | ResBlock Down 128 | Self Attention 128 |
| ResBlock Down 256 | ResBlock Down 256 | ResBlock 128 | ResBlock Down 128 |
| 2 × ResBlock 256 | ResBlock 256 | ResBlock Down 256 | ResBlock Down 256 |
| ResBlock Down 256 | ResBlock Down 256 | ResBlock 256 | Global Mean Pooling |
| 2 × ResBlock 256 | ResBlock 256 | ResBlock Down 256 | Dense → 1 |
| Global Mean Pooling | Global Mean Pooling | ResBlock 256 | |
| Dense → 1 | Dense → 1 | Global Mean Pooling | |
| | | Dense → 1 | |

## G    EXPERIMENTAL DETAILS

**Unconditional Image Generation.**    For all datasets, we perform random horizontal flip as a form of data augmentation. For CelebA, we center crop the image to 140×140 before resizing to 64×64. For LSUN Church, we resize the image to 128×128 directly. We use the same training hyperparameters across all three datasets, which is as follows. We train all models with 120000 training steps with the Adam optimizer with a batch size of 64. We use a learning rate of $1 \times 10^{-4}$ and decay by a factor of 0.1 at training steps 100000 and 110000. We set the number of gradient flow steps to $K = 100$ at training time and $K = 110$ at test time as discussed in Sec. 5.2, and use a step size $\eta = 3$ and noise factor $\nu = 10^{-2}$. The specific residual architectures are given in Table 3. We update model weights using an exponential moving average (Song & Ermon, 2020) given by $\boldsymbol{\theta}' \leftarrow m\boldsymbol{\theta}' + (1-m)\boldsymbol{\theta}_i$, where $\boldsymbol{\theta}_i$ is the parameters of the model at the $i$-th training step, and $\boldsymbol{\theta}'$ is an independent copy of the parameters that we save and use for evaluation. We set $m = 0.998$. We use the LeakyReLU activation with a negative slope of 0.2. We experimented with spectral normalization and self attention layers for unconditional image generation, but found that training was stable enough such that they were not worth the added computational cost. The FID results in Table 1 are obtained by generating 50000 images from the data-dependent prior, and testing the results against the training set for both CIFAR10 and CelebA.

**Conditional Generation with Robust Classifier.**    The unconditional model used for conditional generation is the same model obtained from the section above. The pretrained robust classifier checkpoint is obtained from the `robustness`[3] Python library (Engstrom et al., 2019). It is based on a ResNet50 architecture and is trained with $L_2$-norm perturbations of $\varepsilon = 1$.

We choose the LR-KL variant for our results in Fig. 4a. This means that our conditional gradient flow is given by

$$\mathbf{x}_{k+1} = \mathbf{x}_k - 2\alpha \nabla_\mathbf{x} \log\left( r_\theta(\mathbf{x}_k) * \frac{1}{N} p(y = n|\mathbf{x}_k)^{-1} \right) + \nu \boldsymbol{\xi}_k \tag{32}$$

$$= \mathbf{x}_k - 2\alpha \nabla_\mathbf{x} \left( \log r_\theta(\mathbf{x}_k) - \phi \log p(y = n|\mathbf{x}_k) \right) + \nu \boldsymbol{\xi}_k \tag{33}$$

where in the second line we introduce $\phi$ as a parameter that scales the magnitude of the classifier's gradients so they are comparable to the magnitude of DGGF's gradients. We use $\phi = 0.1$.

**Unpaired Image-to-image Translation.**    The Cat2dog dataset contains 871 Birman cat images and 1364 Samoyed and Husky dog images. 100 of each are set aside as test images. We first resize the images to 84×84 before center cropping to 64×64. Due to the relatively small size of the dataset, we use a shallower residual architecture as compared to CelebA $64^2$ despite the same resolution (Table 3) to prevent overfitting. We also utilize spectral normalization and a self attention layer at the 128-channel level to further boost stability. We set $K = 100$ during training and $K = 110$

---

[3]https://github.com/MadryLab/robustness

Table 4: FID scores for image-to-image translation with the cat2dog dataset.

| Model | FID $\downarrow$ |
|---|---|
| DGGF | 108.10 |
| CycleGAN | 51.79 |

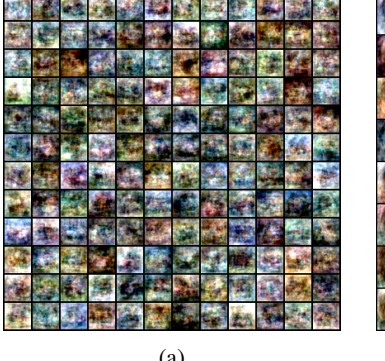 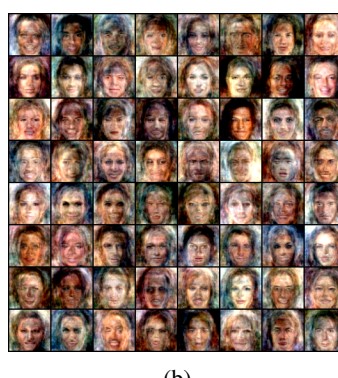 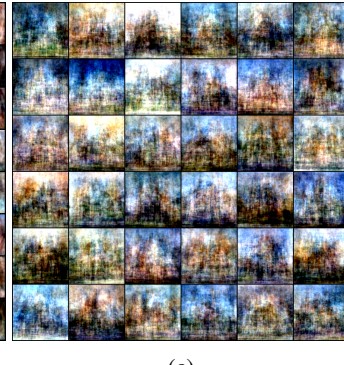

(a)             (b)             (c)

Figure 8: Samples drawn from the data-dependent priors of (a) CIFAR10 $32^2$, (b) CelebA $64^2$ and (c) LSUN Church $128^2$.

at test time. As we observed the model tends to diverge late in training, we limit the number of training steps to 40000, with a decay factor of 0.1 applied to the learning rate at steps 20000 and 30000. All other hyperparameters are kept identical to the experiments on unconditional image generation. We report results for LSIF-Pearson, although we have experimented with LR-KL and found performance to be similar. The FID result in Table. 4 is obtained by translating the 100 test cat images, and testing the results against the 100 test dog images.

# H NEAREST NEIGHBORS

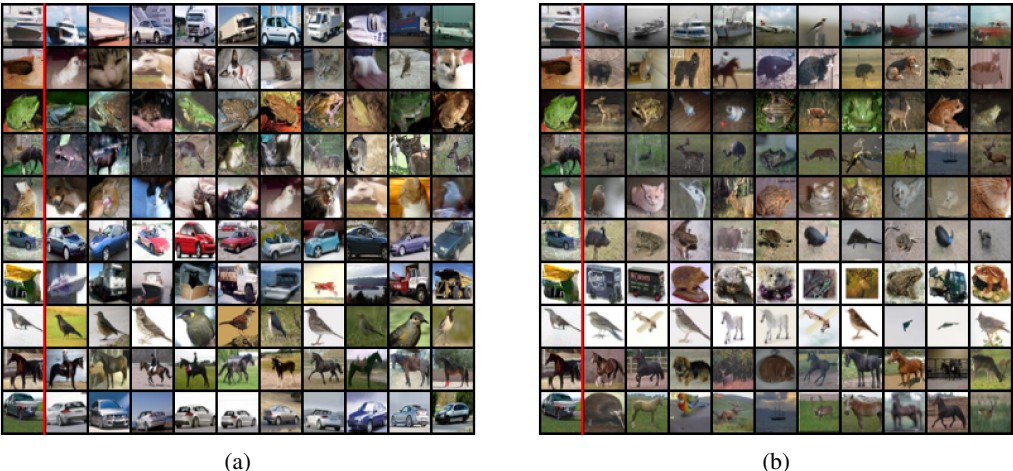

(a)                                         (b)

Figure 9: Nearest neighbor images for CIFAR10 as measured by $L_2$ distance in (a) the feature space of an Inception V3 network pretrained on ImageNet and (b) data space. The column to the left of the red line are samples from DGGF LSIF-$\chi^2$. The images to the right of the line are the 10 nearest neighbors in the training dataset.

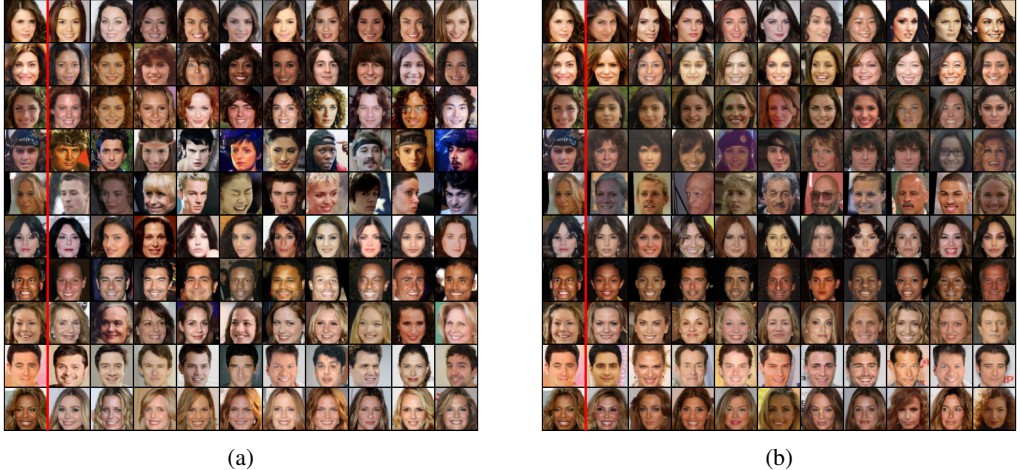

(a)                                         (b)

Figure 10: Nearest neighbor images for CelebA as measured by $L_2$ distance in (a) the feature space of an Inception V3 network pretrained on ImageNet and (b) data space. The column to the left of the red line are samples from DGGF LSIF-Pearson. The images to the right of the line are the 10 nearest neighbors in the training dataset.

## I    UNCURATED SAMPLES DGGF-DDP

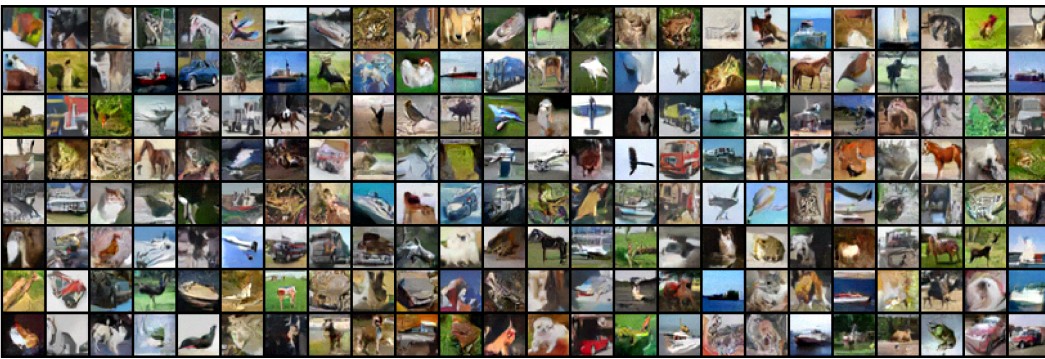

Figure 11: Uncurated samples of CIFAR10 LSIF-Pearson.

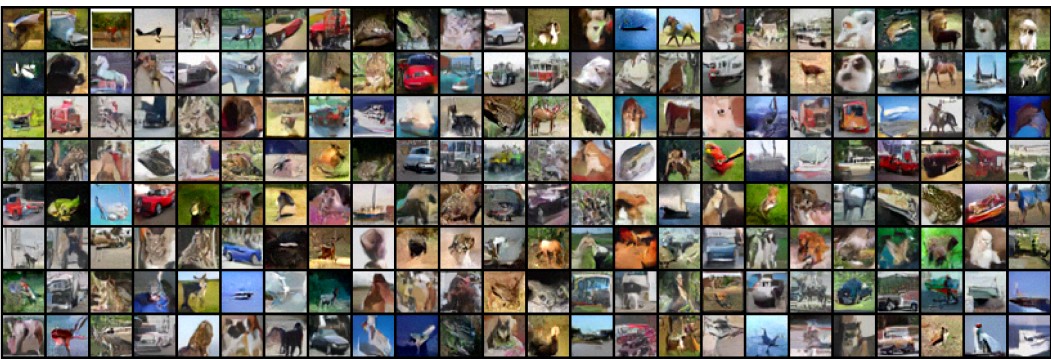

Figure 12: Uncurated samples of CIFAR10 LR-KL.

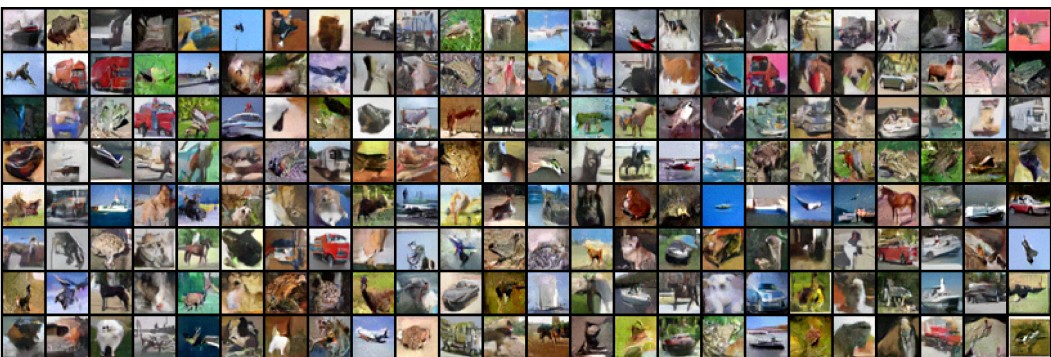

Figure 13: Uncurated samples of CIFAR10 LR-JS.

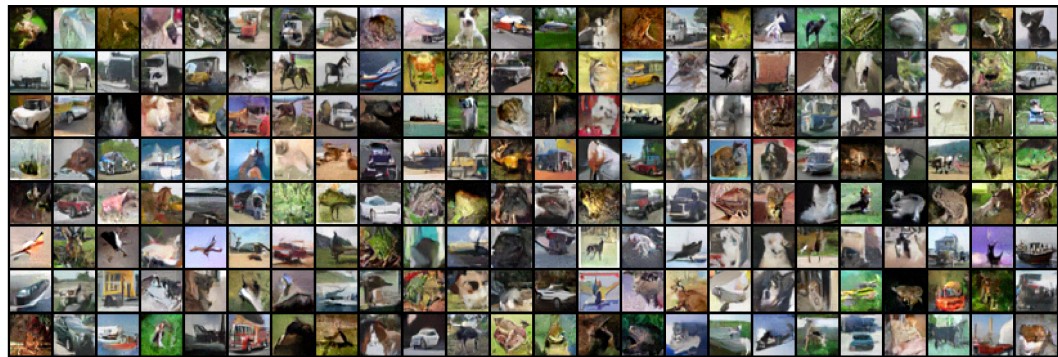

Figure 14: Uncurated samples of CIFAR10 LR-logD.

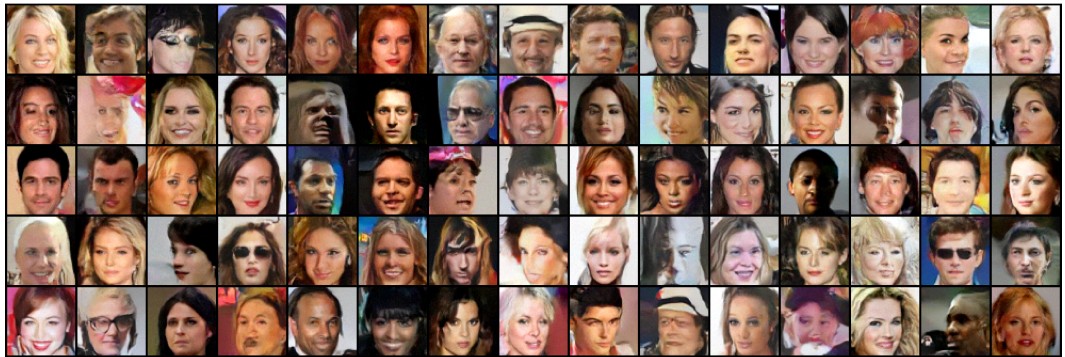

Figure 15: Uncurated samples of CelebA LSIF-Pearson.

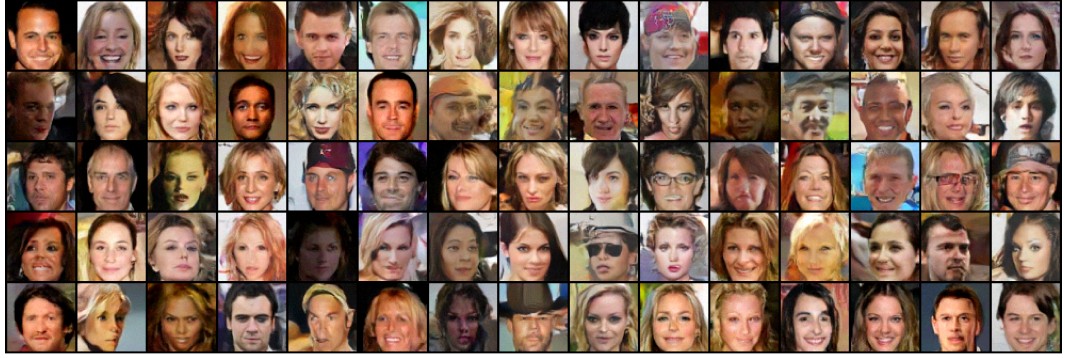

Figure 16: Uncurated samples of CelebA LR-KL.

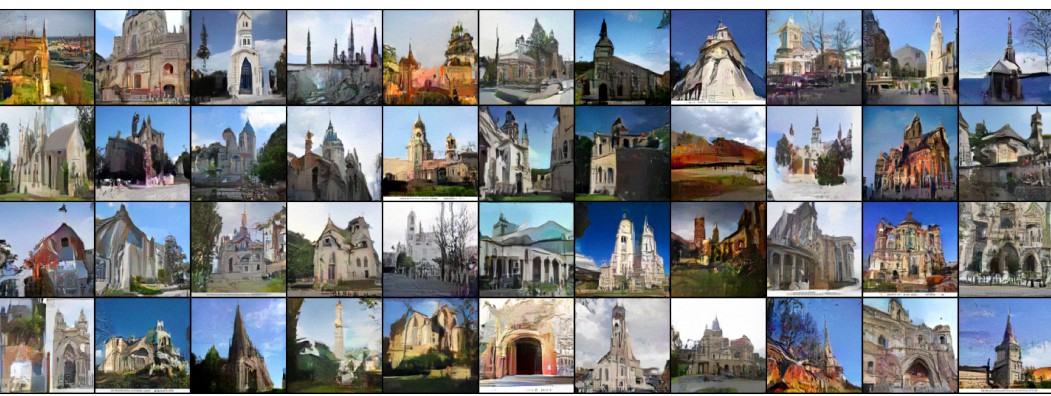

Figure 17: Uncurated samples of LSUN Church LSIF-Pearson.

## J    UNCURATED SAMPLES DGGF-UP

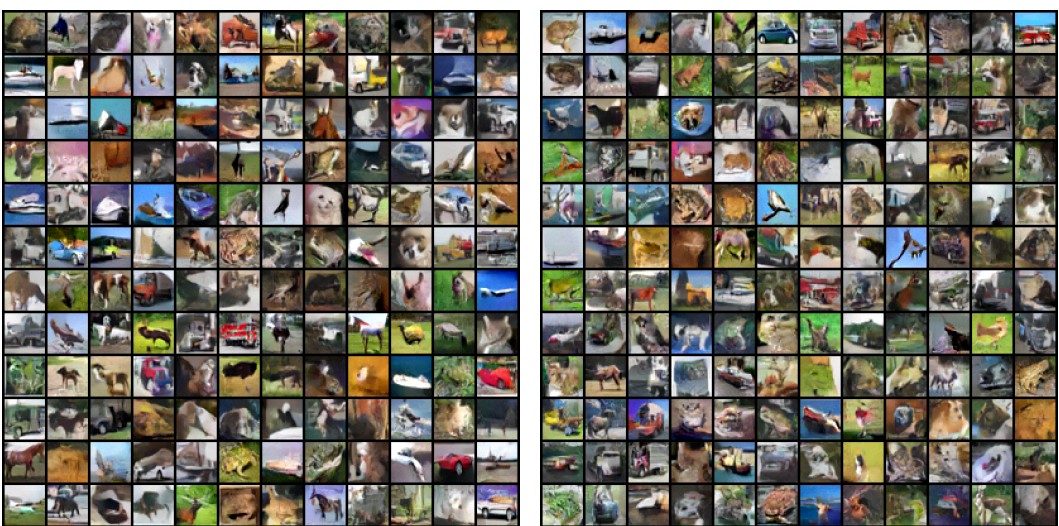

(a) LSIF-Pearson uniform prior.                    (b) LR-KL uniform prior.

Figure 18: Uncurated CIFAR10 samples with DGGF-UP.

