# OpenReview forum: "Deep Generative Wasserstein Gradient Flows"
_ICLR.cc/2023/Conference — Submitted to ICLR 2023_

### Official Review · Reviewer_YDPG · 2022-10-21

**Confidence:** 4
**Correctness:** 2
**Technical Novelty And Significance:** 2
**Empirical Novelty And Significance:** 3
**Recommendation:** 5

**Clarity, Quality, Novelty And Reproducibility:**

**Clarity:** The paper is over-all well-written.
* The link to the article [1] seems to be missed (it introduces density ratio fitting under Bregman divergence):

[1] Sugiyama et. al. Density-ratio matching under the Bregman divergence: a unified framework of density-ratio estimation https://doi.org/10.1007/s10463-011-0343-8

What is `stop_gradient` procedure in the Alorithm 1 in the appendix?
Some hrefs in the paper don’t work (do not redirect to the bibliography section). For example, Gao et. al., 2022. Song and Ermon et. al., 2019

**Quality and Novelty:** The idea of utilizing Wasserstein gradient flows (in particular, with entropy regularized $f$-divergence) is not new, therefore the main contribution the authors present is methodological. In spite of the proposed approach seems to work, there are some problems with theoretical interpretability listed in the weaknesses section. Therefore:
* The questions raised in the weaknesses section should be answered. Probably it is worth considering some toy scenarios with known density ratio function or give up Wasserstein gradient flows interpretation in favor of some other theoretical basis.
* Minor comment, but yet: please add the comparison with JKO-flow approach [2]. This related method seems to surpass the proposed one at CIFAR10 image generation problem.
[2] Fan et. al. Variational Wasserstein gradient flow, https://arxiv.org/pdf/2112.02424.pdf

**Reproducibility:** I didn’t run the code provided but it seems to be clearly written and I have no special doubts regarding reproducibility.


**Strength And Weaknesses:**

**Strength**

* The paper provides a comprehensive list of practical applications and shows practical advantages of the method in several use cases. For example, the proposed density-ratio based method could be adapted for conditional generation on-the-fly.
* DGGF approach seems to overcome the alternative methods based on density ratio function (EPT) and doesn’t require additional latent space to data space generators. The proposed Algorithm is easy to implement (compared to JKO-based Wasserstein gradient flows methods)

**Weaknesses**
* My main concern regarding the proposed method is the lack of theoretical interpretability. The theory of Wasserstein gradient flows is nice but it is not clear if the Algorithm 1 is indeed turning the function $r_{\theta}$ to be the density ratio function $q_t(x)/p(x)$. At first, $r_{\theta}$ does not depend on time which limits the interpretability from a theoretical point of view. There are cases when the prior distribution $q_0(x)$ is significantly different from target distribution $p(x)$ and one can expect that the density ratio function significantly changes along the gradient flow (for example, when the uniform distribution is used as the prior). Secondly, the convergence analysis of the Algorithm 1 itself is also unclear. Bregman divergence optimization is anticipated by stochastic procedure (EM simulation) and it is hard to see if the optimization converges to something relevant. Therefore, special attention should be paid to the convergence analysis of the Algorithm 1 and what is the physical meaning of $r_{\theta^*}$ it converges to. How does $r_{\theta^*}$ resemble the real density ratio function? How does it depend on $K$?
* In light of my previous point I have some doubts concerning the  interpretation of  “Model Estimate of KL over Flow” section. The Figure 4 shows that the quantity $E_{x_{t_k}} \log r_{\theta}(x_{t_k})$ is indeed converges to zero, but if we can interpret the charts as KL convergence remains unclear for me.

**Summary Of The Paper:**

The paper under consideration proposes a method (DGGF) for solving generative modeling related problems. The method is based on Wasserstein gradient flows with potential function given by entropy-regularized $f$-divergence. Essentially, the authors try to approximate density ratio function (DRE) via Bregman divergence combined with Euler-Maruyama simulation. The density ratio function is used at the inference stage allowing a practitioner to sample from target distribution.

**Summary Of The Review:**

In spite of practical applicability of the proposed DGGF approach the theoretical interpretability should be significantly improved. For now there is a gap between Wasserstein gradient flows theory and proposed methodology.

---

> ### Author Response · Authors · 2022-11-15
> **Author's response to reviewer part 1/2**
>
> We thank the reviewer for recognizing the strong practical benefits and simplicity of our approach.
>
> **Comment** -  *“… $r_\theta(x)$  does not depend on time which limits the interpretability from a theoretical point of view. There are cases when the prior distribution is significantly different from target distribution and one can expect that the density ratio function significantly changes along the gradient flow…”*
> and
> *“..physical meaning of $r_\theta$ it converges to. How does $r_{\theta^\*}$ resemble the real density ratio function?”*
> **Response** - We have revised Section 3.1 to improve the clarity of our formalism. To clarify a potential misunderstanding, although our formalism is time-independent, our estimate is not $q_0(x)/p(x)$. $q_0(x)$ and $p(x)$ differ too greatly (i.e., a large density chasm) that a density ratio estimator trained that way would learn a trivial boundary rather than a valid density ratio, as described in Sec. 3.2. Rather, our formalism is effectively estimating a stale ratio $q_K(x)/p(x)$, similar to Ansari et al [3], where $q_K(x)$ is the marginal distribution of particles flowing for $K$ steps. This serves as an approximation to the true gradient flow which requires a time-dependent ratio, which in early experiments we found to lead to unstable and poor training.
>
> In our early experiments, we employed time embeddings to learn a time-dependent DRE; however, this led to unstable training and poor results. We hypothesize that the model does not receive sufficiently stable learning signals as the $x_{1:K}$ change as parameters $r_\theta$ changes, leading to poor performance and unstable training, and leave further exploration on time-dependent DRE to future work. We emphasize this in Sec 3.1. Despite this, our model does not collapse to a single density ratio, but has learnt a varying density ratio over the flow. This is evident in Fig. 4, where the expectation of logR is not constant, but decreases monotonically along the flow. Empirical evidence that our model has learnt a valid density ratio over the flow can also be seen from our Swissroll toy examples in Fig 7 of the appendix, where the results converge well to the training data, as well as our high quality image samples, despite the lack of explicit time embeddings.
>
>
> **Comment** - *“...some doubts concerning the interpretation of “Model Estimate of KL over Flow” section. The Figure 4 shows that the quantity is indeed converges to zero, but if we can interpret the charts as KL convergence remains unclear for me.”*
> **Response** - We have removed the strong claim that $\widehat{D}_{KL} = 0$ can be interpreted as convergence of the samples to the dataset, as we agree that convergence in this sense is much harder to prove. We would like to stress that the KL estimate serves as an interpretation, where we emphasize the relative magnitudes between the data-dependent prior vs uniform prior, and the monotonically decreasing value of $\log r(x)$ over the flow as evidence that our model has learnt a meaningful estimate over the flow. We observe that sample quality, which improves over the flow, is directly correlated to the decreasing KL estimate, which effectively allows the model to be a diagnostic of its own samples.
>
> **Comment** - *“The link to the article [1] seems to be missed”*
> Response - Thank you for the reference [1] and we have included it in the revision. This reference actually contains similar content to the relevant chapter of the density ratio textbook by the same authors, which we have cited previously [2].

---

> > ### Author Response · Authors · 2022-11-15
> > **Author's response to reviewer part 2/2**
> >
> > **Comments** - *“What is stop_gradient procedure in the Algorithm 1 in the appendix?”*
> > Response - The stop-gradient operator refers to disconnecting the tensor $x_K$ from the computation graph that deep learning frameworks (e.g. PyTorch) automatically construct. This prevents backpropagation through the gradient flow, which would be computationally expensive (and unnecessary) as it requires unrolling the computation graph $K$ times.
> >
> > **Comment** - *“Some hrefs in the paper don’t work (do not redirect to the bibliography section). For example, Gao et. al., 2022. Song and Ermon et. al., 2019”*
> > **Response** - Thank you for pointing this out. We have looked through the hrefs and did not find any issues with the aforementioned references. We would appreciate it if the reviewer could refer us to the specific section/page where the hrefs were not redirecting as intended.
> >
> > **Comment** - *“please add the comparison with JKO-flow approach [2].”*
> > **Response** - We have included a comparison with JKO Flow [2] on the CIFAR10 dataset . They achieve an FID of **23.7**. In our recent experiments with larger networks, our preliminary experiments show that for LS-JS, our scores improved from **29.92** to **23.97**, matching JKO-Flow’s performance. The experiments for other variants of DGGF are still ongoing and will report them in the final revision. Notably, JKO-Flow uses a dual network formalism, where the $T$ network is a UNet and $h$ network is a ResNet, while we can match their results using only a single ResNet.
> >
> > We thank the reviewer again for recognizing the strengths of our work, and for their valuable feedback. We request the reviewer to reconsider their score, if our responses have addressed their questions and concerns satisfactorily.
> >
> > [1] Sugiyama et. al. Density-ratio matching under the Bregman divergence: a unified framework of density-ratio estimation https://doi.org/10.1007/s10463-011-0343-8
> > [2] Masashi Sugiyama, Taiji Suzuki, and Takafumi Kanamori. Density ratio estimation in machine learning. Cambridge University Press, 2012a.
> > [3] Refining deep generative models via discriminator gradient flow. In ICLR, 2021.

---

> > > ### Comment · Reviewer_YDPG · 2022-11-19
> > > **Response to the authors' comments**
> > >
> > > I thank the authors for the detailed response. The minor issues/misunderstandings with citations/hrefs are now fixed. Probably, the problem with hrefs was caused by my local laptop. Now let’s move on to the main disputed points that I still see in the work.
> > >
> > > 1. I still have doubts regarding the interpretation of  “Model Estimate of KL over Flow” section. In the current version of the manuscript you state, that $E_{x_{k}}\log r_{\theta}(x_k)=\int q(x_k) \log(q/p)(x_k) d x_k$. I don’t understand this formula at all. Even if the density ratio $r_{\theta}$ you model is indeed a density ratio between some pdf $q$ and target data pdf $p$ the distribution (say, $q_k$) of the samples $x_k$ doesn’t coincide with $q$. I want to say, that the correct formula is $E_{x_k}\log r_{\theta}(x_k) = \int q_k(x_k) \log(q/p)(x_k) d x_k$ and I don’t know how to interpret this formula as a KL divergence.
> > >
> > > 2. More general and important dispute point. Thank you for clarifying section 3.1, but I still don’t understand what $r_{\theta^\star}$ actually models. What the physical meaning of $r_{\theta^\star}$? For now there are no theoretical foundations for the validity of your approach, you have only hypotheses. And I believe, that the theoretical background behind your empirical results is no less important than the empirical results itself.
> > >
> > > Based on the aforementioned comments, I consider leaving my score unchanged.

---

> > > > ### Author Response · Authors · 2022-11-30
> > > > **Author's response to reviewer**
> > > >
> > > > We thank the reviewer for their follow up comments.
> > > >
> > > > **KL estimate** - Upon further consideration, we agree with the reviewer that because of our use of a ‘stale’ density ratio estimate, it is not accurate to call it a valid KL estimate over the flow, although empirically it seems to behave as such. We will remove the relevant section from our final revision of the paper. As the KL estimate is not the main result, we do not believe its removal reduces the impact and novelty of our paper.
> > > >
> > > > **Meaning of $r_\theta$ and of the gradient flow** - We would like to provide further explanations of what $r_\theta$ represents, and more generally what our gradient flow is simulating.
> > > >
> > > > First let’s try to understand $r_\theta$ at convergence. Consider what happens at convergence during training. At the final training step $N$, we sample $x_0 \sim q_0(x)$ and simulate our time-independent gradient flow with $r_{N-1}(x)$, where $r_{N-1}$ denotes the estimator that has converged in training step $N-1$. We denote the samples as drawn from the distribution $q^K_{N-1}(x) = \int q_0(x’) M_{N-1}(x|x’) dx’$, where $M_{N-1}$ denotes the transition kernel of the flow using $r_{N-1}$. We optimize the Bregman divergence using the $x^K_{N-1} \sim q^K_{N-1}$ and $x \sim p(x)$. Assuming convergence of the BD, the density ratio estimator we obtain at the end of training is $r_{\theta*}(x) = r_N(x) = q^K_{N-1} / p$.
> > > >
> > > > At test time, we now simulate the gradient flow given by
> > > > \begin{align} dx_t &= -\nabla_x f'(r_N(x))dt + \sqrt{2\gamma}dW_t \\\ &= -\nabla_x f'(q^K_{N-1}(x)/p(x))dt + \sqrt{2\gamma}dW_t \end{align}
> > > >
> > > > We can understand what this flow is doing when we choose $f’=\log$, which corresponds to KL divergence.
> > > >
> > > > \begin{align} dx_t&= -\nabla_x \log(q^K_{N-1}(x)/p(x))dt + \sqrt{2\gamma}dW_t \\\ &= -\nabla_x \log(q_t(x)/p(x))dt + \nabla_x \log(q_t(x)/q^K_{N-1}(x))dt + \sqrt{2\gamma}dW_t \end{align}
> > > >
> > > > This resultant flow corresponds to the FPE that minimizes the following functional
> > > >
> > > > \begin{equation}
> > > > \mathcal{F}(q_t) = KL(q_t || p) - KL(q_t || q^K_{N-1}) + \gamma H(q_t)
> > > > \end{equation}
> > > >
> > > > Thus, our time-independent gradient flow ‘cancels’ out the tricky $q_t$ term such that we don’t have to deal with time-dependence. The interpretation is that since we are minimizing this functional, the *implicit* $q_t$ of the gradient flow is pushed towards $p(x)$ by the first KL term, whilst away from $ q^K_{N-1}$ in the second term, which is the marginal distribution of samples from flowing during the *last step* of training. Empirically, we see that at test time, the samples are high-quality and indeed close to $p(x)$. While this analysis works for KL, we find that interestingly the other $f$ divergences also produce comparable visual results, which we think is worthy of future investigations.

---

> > > > > ### Comment · Reviewer_YDPG · 2022-12-01
> > > > > **Response to the authors**
> > > > >
> > > > > I thank the authors for the attempt to explain what $r_{\theta}$ means. Given your explanation for $KL$ divergence, the theoretical justification of your approach is still questionable. If the FPE you model is actually minimizes (for $KL$ divergence case) the functional:
> > > > > $$
> > > > > \mathcal{F}(q_t) = KL(q_t \Vert p) - KL(q_t \Vert q_{N - 1}^K) + \gamma H(q_t)
> > > > > $$
> > > > > then the resulting distribution is actually biased (because of the term $- KL(q_t \Vert q_{N - 1}^K)$) and the effect of this bias is unclear. For me the paper under consideration still lacks a theoretical explanation. Based on this I leave my score unchanged.

---

### Official Review · Reviewer_NwTe · 2022-10-24

**Confidence:** 4
**Correctness:** 3
**Technical Novelty And Significance:** 2
**Empirical Novelty And Significance:** 2
**Recommendation:** 3

**Clarity, Quality, Novelty And Reproducibility:**

It is suggested to provide a pseudocode of the training algorithm in the main body instead of the appendix. If possible, please also include a sampling algorithm for reference.
 The results might be important for developing deep generative models if the correctness can be justified.

**Strength And Weaknesses:**

The work is not theoretically sound, and the proposed algorithm lacks justifications of its correctness both intuitively and technically.
It is well-known that Wasserstein gradient flows can be utilized in deep generative modeling. There are a series of publications that have introduced Wasserstein gradient flows in deep generative modeling and explored their potential in analyzing or developing deep generative modeling algorithms, for example please see [1-9]. Most of these relevant papers are not mentioned in the related work. It is not novel to leverage the Wasserstein gradient flow of the entropy regularized f-divergence in deep generative modeling and to further simulate the gradient flow with a density ratio estimation procedure [8, 9]. The empirical evaluation of the proposed method seems reasonable.

The central part of the proposed method is to numerically simulate the relevant Wasserstein gradient flows with density ratio estimation between the pushforward distribution $q_t$ and the target distribution $p$. According to the authors on page 4, “The simulation of Eq. 8 requires a time-dependent estimate of the density ratio; however, we train the density ratio estimator $r_\theta$ without explicit dependence on time”. As far as I’m concerned, the numerical simulation formula in Eq. 8 should explicitly depend on a time-dependent density ratio which is a basic result of Euler-Maruyama discretization. It is hard to capture the basic intuition of recasting this density ratio as a time-independent one, especially from a perspective of simulating a time-dependent Wasserstein gradient flow path. In practice, the proposed method does only estimate a time-independent density ratio and plugs it in Eq. 8 as indicated by the pseudocode of the training algorithm in the appendix. Furthermore, a natural question would be raised regarding the convergence properties of the proposed algorithm. Suppose that the training of the proposed model is completed, the pushforward distribution $q_{t_K}$ is approximately equal to the target distribution $p$ when $t_K$ is large enough. Accordingly, the estimated density ratio between $q_{t_K}$ and $t_K$ would be expected to approximate a constant 1 almost everywhere. This observation is basically inspired by the convergence behavior of Wasserstein gradient flows. Then a straightforward question is how to generate new samples at test time just using a time-independent density ratio while the estimated density ratio is approximately equal to 1. Due to this possible issue, it is recommended to justify the convergence property of the iterations in Eq. 8 when a claimed time-independent density ratio estimator is deployed.

[1] Geometrical Insights for Implicit Generative Modeling, 2017.
[2] Sliced-Wasserstein Flows: Nonparametric Generative Modeling via Optimal Transport and Diffusions, ICML 2019.
[3] Sobolev Descent, AISTATS 2019.
[4] Maximum mean discrepancy gradient flow, NeurIPS 2019.
[5] Deep generative learning via variational gradient flow, ICML 2019.
[6] Refining deep generative models via discriminator gradient flow, ICLR 2021.
[7] On the convergence of gradient descent in GANs: MMD GAN as a gradient flow, AISTATS 2021.
[8] Reﬁning deep generative models via discriminator gradient ﬂow, ICLR 2021.
[9] Deep generative learning via Euler particle transport, MSML 2022.


**Summary Of The Paper:**

This work re-explores deep generative modeling by minimizing the entropy regularized f-divergence in the Wasserstein-2 space of probability measures and derives an algorithm based on an ambiguous discretization method of the relevant Wasserstein gradient flows. Some experimental results on image synthesis are presented to support the proposed method.

**Summary Of The Review:**

The paper lacks of both theoritical and computational justification.

---

> ### Author Response · Authors · 2022-11-15
> **Author's response to reviewer part 1/2**
>
> We thank the reviewer for their insightful comments and for acknowledging the experimental results of our work.
>
> **Comment** - *“...series of publications that have introduced Wasserstein gradient flows in deep generative modeling and explored their potential in analyzing or developing deep generative modeling algorithms, for example please see [1-9]. Most of these relevant papers are not mentioned in the related work.”*
> **Response** - Thank you for the references. We have already cited [5], [6] and [8] previously and have included relevant citations for [2], [3], [4] and [7] under related works in our revision. [1] discusses the various divergences and how they relate to the adversarial generative framework (GANs). They additionally cover geometrical properties of the various objectives. There are no explicit connections to gradient flows as far as we are aware, so we did not cite this in our paper. While the geometrical insights could have connections to WGFs, we leave such exploration to future studies.
>
> We reiterate that we do not claim that generative modeling with f-divergence WGFs are novel, and we have discussed prior studies [6, 8] extensively in related works. However, our model greatly improves on [6, 8]. [6] tackles the problem of sample refinement with GAN discriminators, whereas we generalize the work to sample generation from scratch. [8] proposes to train an auxiliary generator as they were unable to scale their WGF directly to the data space. Furthermore, the density ratio estimator in [8] suffers from the issue where the ratio collapses to 1 and no useful gradients can be obtained at test time, which the reviewer has pointed out (hence the use of an auxiliary generator). DGGF resolves both these issues: 1) we perform gradient flow in data space with a single network, 2) our density ratio estimator does *not* collapse to 1, as we elaborate in greater detail below.
>
> **Comment** - *"It is suggested to provide a pseudocode of the training algorithm in the main body instead of the appendix. If possible, please also include a sampling algorithm for reference."*
> **Response** - We have included training and sampling pseudocode in the main text as suggested.
>
> **Comment** - *“...numerical simulation formula in Eq. 8 should explicitly depend on a time-dependent density ratio... It is hard to capture the basic intuition of recasting this density ratio as a time-independent one…”*
> and
> *“...the estimated density ratio between and would be expected to approximate a constant 1 almost everywhere…a straightforward question is how to generate new samples at test time just using a time-independent density ratio while the estimated density ratio is approximately equal to 1.*
> **Response** - We have revised section 3.1 so that our formalism is clearer. Our formalism is effectively estimating a stale ratio $q_K(x)/p(x)$, similar to Ansari et al [6], where $q_K(x)$ is the marginal distribution of particles flowing for K steps. This serves as an approximation to the true gradient flow which requires a time-dependent ratio.

---

> > ### Author Response · Authors · 2022-11-15
> > **Author's response to reviewer part 2/2**
> >
> > In our early experiments, we employed time embeddings to learn a time-dependent DRE. However, this led to poor results. We hypothesize that the model did not receive sufficiently stable learning signals as the $x_{1:K}$ change as parameters of $r_\theta$ changes, leading to poor performance and unstable training. We emphasize this in Sec 3.1. In contrast, using a *time-independent* DRE significantly improved results, to the point that we were able to generate 128x128 images.
> >
> > This result is intriguing and relevant to the generative modeling community at large; as the reviewers point out, the conventional thinking is that time-dependency is necessary. But our results show that despite using a DRE without explicit dependence on time, our model does *not* collapse to a single density ratio with zero gradients at the end of training. This is evident in Fig. 4, where the expectation of logR is not constant, but decreases monotonically along the flow. We posit that our DRE has learnt to associate samples with distributions along the flow. Empirical evidence that our model has learnt a valid density ratio over the flow can also be seen from our Swissroll toy examples in Fig 7 of the appendix, where the results converge well to the training data, as well as our high quality image samples.
> >
> > The main novelty of our paper is primarily to showcase the interesting practical benefits of WGF methods, such as high-quality, high-resolution samples, density ratio composition and direct application to unpaired image translation, all of which have not been the focus of prior works. Our model proposes to approximate the time-dependent density ratio without explicit dependence on time, and we find that this works surprisingly well in practice. We believe that these results are relevant to the generative modeling community, and request the reviewer to evaluate the paper along these practical lines. We thank the reviewer again for their valuable feedback and request that the reviewer consider their score if possible.
> >
> > [1] Geometrical Insights for Implicit Generative Modeling, 2017.
> > [2] Sliced-Wasserstein Flows: Nonparametric Generative Modeling via Optimal Transport and Diffusions, ICML 2019.
> > [3] Sobolev Descent, AISTATS 2019.
> > [4] Maximum mean discrepancy gradient flow, NeurIPS 2019.
> > [5] Deep generative learning via variational gradient flow, ICML 2019.
> > [6] Refining deep generative models via discriminator gradient flow, ICLR 2021.
> > [7] On the convergence of gradient descent in GANs: MMD GAN as a gradient flow, AISTATS 2021.
> > [8] Deep generative learning via Euler particle transport, MSML 2022.

---

### Official Review · Reviewer_yo3y · 2022-10-25

**Confidence:** 2
**Correctness:** 3
**Technical Novelty And Significance:** 3
**Empirical Novelty And Significance:** 2
**Recommendation:** 5

**Clarity, Quality, Novelty And Reproducibility:**

The submission is concise and of excellent quality. The submission consists of code.

**Strength And Weaknesses:**

Strengths
* [S1]: The paper addresses an important problem in an interesting way, based on a clear motivation.
* [S2]: The method is sensible and clearly presented, I enjoyed reading it.

Weaknesses
*[W1] Insufficient empirical evidence; several experimental comparisons are absent and should be included (see below for further details).


Details:
* Experiments(baselines):  Considering there are few related work that addressing the intractable chanlledge, some baseline method([1][2]) should be included for comparision.
* Can you further investigate the algorithm's computational complexity considering the training is ? Is it possible to quantify the performance on larger data sets?
* Can a pseudo-code be added to the training/inference procedures?
* Can a discussion be included to this concurrent work? https://arxiv.org/pdf/2209.11178v4.pdf

[1] Large-scale wasserstein gradient flows.
[2] Variational wasserstein gradient flows.

**Summary Of The Paper:**

This paper explores a unconditional learning generation model using Wasserstein Gradient Flow directly on the data space. To make the training feasible, the authors propose to use a deep density ratio estimator for learning in Wasserstein space.Authors demonstrate the proposed method on Cifar10 and CelebA, and the result shows that DGGF is able to generate high-quality images.

From a conceptual point of view, the work differs from related approaches, in the sense that it tackles the problem of learning unconditional generative models directly on data space. However, from computational point of view, I found it quite simililar to Energy-based models.

**Summary Of The Review:**

Overall, I tend to give a negative vote, but the score can still be adjusted if my concerns can be addressed. My main concern comes from the fact that the experimental results of the article are not strong, and there are still many aspects that could be discussed quantitatively (e.g. on a larger datasets, reporting the training speed of the algorithm, etc.), which makes the strengths of WGGF unclear to me.

---

> ### Author Response · Authors · 2022-11-15
> **Author's response to reviewer**
>
> We thank the reviewer for recognizing the impacts of our work and for recognizing its motivations.
>
> **Comment** - *“Considering there are few related work that addressing the intractable challenge, some baseline method([1][2]) should be included for comparison.”*
> **Response** - We did not make comparisons with [1] as it does not include image experiments, possibly due to constraints of the input convex neural networks used to approximate the JKO scheme in their formulation. We have included a comparison with JKO Flow [2] on the CIFAR10 dataset . They achieve an FID of **23.7**. In our recent experiments with larger networks, our preliminary experiments show that for LS-JS, our scores improved from **29.92** to **23.97**, matching JKO-Flow’s performance. The experiments for other variants of DGGF are still ongoing and will report them in the final revision. Notably, JKO-Flow uses a dual network formalism, where the $T$ network is a UNet and $h$ network is a ResNet, while we can match their results using only a single ResNet.
>
> **Comment** - *"Can you further investigate the algorithm's computational complexity considering the training is ? Is it possible to quantify the performance on larger data sets?"*
> and *"Can a pseudo-code be added to the training/inference procedures?"*
> **Response** - We have included the training and sampling pseudocode in the main text. The training complexity is roughly $O(KNH)$, where $K$ is the number of flow steps, $N$ is the number of training steps and $H$ is the size of the network. We have investigated generative modeling of datasets with images up to 128x128, which we believe is the highest dimensional dataset that have been explored in any generative gradient flow works. Due to compute constraints, we are not able to extend our studies to large datasets like ImageNet.
>
> **Comment** - *“Can a discussion be included to this concurrent work? https://arxiv.org/pdf/2209.11178v4.pdf”*
> **Response** - We thank the reviewer for sharing this interesting paper. The paper proposes to treat images like electric charges, which evolve under the Poisson equation. The solution to the Poisson equation is a Fokker-Planck equation without a noise term, hence an ODE and not a SDE like in our work. The paper then proposes to treat the ODE as a forward process, which is a repulsive process where images are repelled from each other like repulsive charges, then learning a reverse process where samples are attracted back to the data distribution like attractive charges. While the methodology is closer to that of a continuous normalizing flow/diffusion model, we agree that it is an interesting way to interpret gradient flows. We have thus included a reference in our revision.
>
> We thank the reviewer again for recognizing the strengths of our work, and for their valuable feedback. We request the reviewer to reconsider their score, if our responses have addressed their questions and concerns satisfactorily.
>
> [1] Large-scale wasserstein gradient flows. Advances in Neural Information Processing Sys-
> tems, 34:15243–15256, 2021.
> [2] Variational wasserstein gradient flow. arXiv preprint arXiv:2112.02424, 2021.

---

### Author Response · Authors · 2022-11-15
**Author's response to all reviewers**

We thank the reviewers for their time and feedback. The reviewers have noted that our paper addresses an important problem in the gradient flow literature, provides strong practical results and applications, and is easier to implement than other Wasserstein gradient flow methods.

The reviewers have also raised several concerns that we have addressed in detail in our individual responses, as well as in our revised paper where the changes made are in blue.

We provide a summary of our responses to common issues by the reviewers here.

**Comparison to JKO-Flow** - We have included a comparison with JKO Flow [2] on the CIFAR10 dataset . They achieve an FID of **23.7**. In our recent experiments with larger networks, our preliminary experiments show that for LS-JS, our scores improved from **29.92** to **23.97**, matching JKO-Flow’s performance. The experiments for other variants of DGGF are still ongoing and will report them in the final revision. Notably, JKO-Flow uses a dual network formalism, where the $T$ network is a UNet and $h$ network is a ResNet, while we can match their results using only a single ResNet.

**Time-independent density ratio estimate** - Given the reviewer comments, there appears to be a misunderstanding of how our density ratio estimator is trained due to the time-independent approximation of the gradient flow SDE. We have revised Section 3.1  to clarify our formalism: our method estimates a stale ratio $q_K(x)/p(x)$, similar to [3], where $q_K(x)$ is the marginal distribution of particles flowing for K steps. This serves as an approximation to the true gradient flow which requires a time-dependent ratio. This setup was motivated by early experiments with poor and unstable performance when we attempted to train a time-dependent model with $x_{1:K}$. Interestingly, the model learns a varying density ratio over the gradient flow *without* explicit time embeddings, as seen from Fig. 4. Empirically, it is able to generate high quality images at test time and does not suffer from the issue where the model collapses to a single density ratio, unlike the baseline EPT [2].

**Overall contribution** - We would like to emphasize to the reviewers that the main novelty of our paper is primarily to showcase the practical benefits of WGF methods, such as (i) high-quality, high-resolution samples, (ii) density ratio composition and (iii) direct application to unpaired image translation, all of which have not been previously demonstrated. Prior works on WGF methods resort to complex optimization schemes such as the JKO scheme, or relied on auxiliary generators and have failed to scale WGF methods to high-dimensional datasets [2]. To circumvent these issues, we propose to approximate the time-dependent density ratio without *explicit* dependence on time. We find that this works well in practice, which is surprising given that current WGF theory indicates that the DRE should be time-dependent, and we leave theoretical characterizations into why that is so to future work.

**Our work informs the community that explicit time dependence, which is complicated to implement, is not necessary for WGF methods to work well in practice. Our model is simple to implement, yet generates high quality samples and scales to high dimensional datasets. Furthermore, we showcase the unique practical applications of WGFs, which in contrast to alternatives such as diffusion and energy-based models, require no further modifications to the framework. We believe these are significant results for the wider gradient flow community, and serve as motivation for further studies into the adoption of WGFs in generative modelling.**

**We believe that our empirical results are relevant to the generative modeling community and request the reviewers to evaluate the paper along these practical lines.**

[1] Variational wasserstein gradient flow. arXiv preprint arXiv:2112.02424, 2021.
[2] Deep generative learning via euler particle transport. In Mathematical and Scientific Machine Learning, pp. 336–368. PMLR, 2022.
[3] Refining deep generative models via discriminator gradient flow. In ICLR, 2021.

---

### Decision · Program_Chairs · 2023-01-20

**Decision:**

Reject

**Justification For Why Not Higher Score:**

N/A

**Justification For Why Not Lower Score:**

N/A

**Metareview: Summary, Strengths And Weaknesses:**

Thank you for submitting your work to ICLR 2023 and providing a rebuttal and revised version of your work. Unfortunately, there are lingering concerns preventing us from recommending acceptance of this work. The main issue seems to be the justification of Algorithm 1 for training the flow, and whether this procedure indeed produces the desired flow. We encourage the authors to try and address this issue in future revisions of this work.